

# Black and brown carbon over central Amazonia: Long-term aerosol measurements at the ATTO site

Jorge Saturno[1], Bruna A. Holanda[1], Christopher Pöhlker[1], Florian Ditas[1], Qiaoqiao Wang[1,2], Daniel Moran-Zuloaga[1], Joel Brito[3,4], Samara Carbone[3,5], Yafang Cheng[1], Xuguang Chi[6], Jeannine Ditas[1,2], Thorsten Hoffmann[7], Isabella Hrabe de Angelis[1], Tobias Könemann[1], Jošt V. Lavrič[8], Nan Ma[1,2], Jing Ming[1], Hauke Paulsen[9], Mira L. Pöhlker[1], Luciana V. Rizzo[10], Patrick Schlag[3], Hang Su[1], David Walter[1], Stefan Wolff[1], Yuxuan Zhang[1], Paulo Artaxo[3], Ulrich Pöschl[1], and Meinrat O. Andreae[1,11]

[1]Biogeochemistry & Multiphase Chemistry Departments, Max Planck Institute for Chemistry, P. O. Box 3060, 55020 Mainz, Germany.
[2]Jinan University Institute for Environmental and Climate Research, Guangzhou, China.
[3]Department of Applied Physics, Institute of Physics, University of São Paulo (USP), Rua do Matão, Travessa R, 187, CEP 05508-900, São Paulo, SP, Brazil.
[4]Laboratory for Meteorological Physics, Université Clermont Auvergne, Clermont-Ferrand, France.
[5]Institute of Agrarian Sciences, Federal University of Uberlândia, Uberlândia, Minas Gerais, Brazil.
[6]Institute for Climate and Global Change Research & School of Atmospheric Sciences, Nanjing University, Nanjing, 210093, China.
[7]Department of Chemistry, Johannes Gutenberg University, Mainz, Germany.
[8]Department of Biogeochemical Systems, Max Planck Institute for Biogeochemistry, 07701 Jena, Germany.
[9]Institute of General Botany, Johannes Gutenberg University, Mainz, Germany.
[10]Departamento de Ciencias Ambientais, Universidade Federal de Sao Paulo, Diadema, SP, Brasil.
[11]Scripps Institution of Oceanography, University of California San Diego, La Jolla, CA 92098, USA.

*Correspondence to*: Jorge Saturno (j.saturno@mpic.de) and Christopher Pöhlker (c.pohlker@mpic.de)

**Abstract.** The Amazon rain forest is considered a very sensitive ecosystem that could be significantly affected by a changing climate. It is still one of the few places on Earth where the atmosphere in the continent approaches near-pristine conditions for some periods of the year. The Amazon Tall Tower Observatory (ATTO) has been built in central Amazonia to monitor the atmospheric and forest ecosystem conditions. The atmospheric conditions at the ATTO site oscillate between biogenic and biomass burning (BB) dominated states. By using a comprehensive ground-based aerosol measurement setup, we studied the physical and chemical properties of aerosol particles at the ATTO site. This paper presents results from 2012 to 2017, with special focus on light absorbing aerosol particles. The aerosol



absorption wavelength dependence (expressed as the absorption Ångström exponent, $\mathring{a}_{abs}$) was usually

below 1.0 and increased during the presence of smoke transported from fires in the southern and eastern

regions of the Amazon or advected from savanna fires in Africa. In this study, the brown carbon (BrC)

contribution to light absorption at 370 nm was obtained by calculating the theoretical wavelength

dependence of $\mathring{a}_{abs}$ (WDA). Our calculations resulted in BrC contributions of 17 – 29 % (25[th] and 75[th]

percentiles) to total light absorption at 370 nm ($\sigma_{ap\,370}$)  during the measurement period (2012 – 2017).

The BrC contribution increased up to 27 – 47 % during fire events occurring under the influence of El

Niño, between September and November 2015. An extended time series of ATTO and ZF2 (another

Amazon rain forest sampling site) data showed enhanced light scattering and absorption coefficients

during El Niño periods in 2009 and 2015. Long-range transport (LRT) aerosol particles that reached the

central Amazon Basin from Africa or from southern Amazon exhibited a wide range of black carbon

(BC) to carbon monoxide (CO) enhancement ratios ($ER_{BC}$) (between 4 and 15 ng m$^{-3}$ ppb$^{-1}$) reflecting

the variability of fuels, combustion phase, and removal processes in the atmosphere. Higher $ER_{BC}$ were

measured during the dry season when we observed values up to 15 ng m$^{-3}$ ppb$^{-1}$, which were related to

the lowest single scattering albedo ($\omega_0$) measured during the studied period, (0.86 – 0.93). A

parameterization of $\mathring{a}_{abs}$ as a function of the BC to OA mass ratio was investigated and was found

applicable to tropical forest emissions but further investigation is required, especially by segregating

fuel types. Additionally, important enhancements of the BC mass absorption cross-section ($\alpha_{abs}$) were

found over the measurement period. This enhancement could be linked to heavy coating of the BC

aerosol particles. In the future, the BC mixing state should be systematically investigated by using

different instrumental approaches.

**1 Introduction**

Atmospheric aerosol particles affect the Earth's climate through different mechanisms. Direct

mechanisms include the aerosol particle interaction with radiation by scattering and absorption. The

balance between scattering and absorption can lead to warming or cooling of the atmosphere (IPCC,

2013). Moreover, aerosol-cloud interactions related to cloud formation and cloud microphysical



modification, are related to high uncertainties, especially due to the lack of knowledge of pre-industrial

cloud condensation nuclei (CCN) availability (Carslaw et al., 2013) and aerosol particles spatial

distribution in the atmosphere (Andreae, 2007).

Continuous aerosol measurements at remote continental locations are crucial to understand atmospheric

conditions prior to industrialization and reduce the uncertainties in climate models (Seinfeld et al.,

2016). The Amazon Basin is one of the few continental areas in the world where the atmosphere can be

studied in near-pristine conditions during some periods of the year (Andreae et al., 2015). However,

measuring under near-pristine to pristine conditions is quite challenging even in very remote places

because anthropogenic pollution is rather persistent and, thus, reaches almost every continental place on

the planet (Andreae, 2007; Chi et al., 2013; Hamilton et al., 2014). The Amazon rain forest has been

impacted by intensified agriculture and associated deforestation in the southern and eastern areas and

infrastructural development in the last 50 years (Artaxo et al., 2013; Davidson et al., 2012). Given these

circumstances, only when air masses travel over clean marine areas and the rain-related scavenging is

significant, the observations approach near-pristine aerosol particle levels (Andreae et al., 2012, 2015).

Biogenic primary and secondary organic aerosol particles over the Amazon rain forest are ubiquitous

throughout the year (Martin et al., 2010). During the dry season (August – November), when fires are

frequent in the forest and its peripheries, the background biogenic aerosol is overwhelmed by BB

smoke. Despite the occurrence of natural tropical forest fires, most of the fire episodes in the Amazon

rain forest peripheries occur due to human activity, including land use change, brush clearing for

agricultural activities and burning of agricultural waste (Andreae, 1991; Crutzen and Andreae, 1990).

Additionally, cooperative burning of savannas is a common practice by indigenous communities in the

region and it helps to prevent larger wildfires when burned areas can act as "firebreaks" (Bilbao et al.,

2010). Starting in August, the dry season is characterized by aerosol number concentrations of

$1000 - 3000$ cm$^{-3}$ (Andreae et al., 2015). Another characteristic of this period is the abundance of black

carbon (BC). This type of aerosol particles are primarily emitted by flaming and smoldering fires

together with large amounts of organic aerosol (OA) (Andreae and Merlet, 2001) and are considered an

important short-lived climate forcing agent (Bond et al., 2004, 2013). The BC co-emitted light

absorbing fraction of OA is called *brown carbon* (BrC) (Andreae and Gelencsér, 2006). The BC + BrC





aerosol fraction is commonly defined as *light-absorbing carbonaceous matter* (LAC). The mentioned nomenclature is in accordance with the one compiled by Petzold et al. (2013). A list of frequently used

acronyms and symbols can be found in Table 1.

During combustion, aerosol particles are co-emitted with carbon monoxide (CO). The ratio between aerosol mass or number concentrations and CO has been used to trace air masses origin and age (Guyon et al., 2005; Janhäll et al., 2010). Enhancement ratios ($ER_{BC}$) for open biomass burning measured for boreal forest smoldering fires have an average $ER_{BC}$ of 1.7 ng m$^{-3}$ ppb$^{-1}$ (Kondo et al., 2011). On the

other hand, agricultural fires exhibit higher $ER_{BC}$ compared to forest fires, with reported values varying between 2.2 and 29.8 ng m$^{-3}$ ppb$^{-1}$, see Mikhailov et al. (2017) and references therein.

Biomass burning plumes are usually dominated by accumulation mode aerosol particles, which are efficient to scatter radiation and also rich in BC. In the absence of BB aerosol particles, the biological coarse mode particles become dominant in terms of mass and the aerosol optical properties are affected.

Therefore, clear seasonal trends in scattering and absorption have been observed by long-term measurements in the Amazon region (Rizzo et al., 2013).

The light absorption of BC has a wavelength dependence that is conditioned by the BC mixing state, its size distribution and the composition of co-emitted particles (Andreae and Gelencsér, 2006; Kirchstetter et al., 2004; Lack et al., 2013; Schuster et al., 2016). The wavelength dependence is described by the

absorption Ångström exponent ($\mathring{a}_{abs}$) (Ångström, 1929), which can vary from low values ($\mathring{a}_{abs} = 1.0 \pm 0.1$, weak spectral dependence), usually associated to fossil fuel emitted BC (Bond and Bergstrom, 2006), up to high values ($\mathring{a}_{abs} = 6\text{-}7$, strong spectral dependence) for organic-rich aerosol, e.g., humic-like substances (HULIS) (Hoffer et al., 2006). Measurements at an Amazonian forest site during the dry season resulted in $\mathring{a}_{abs}$ average values below 1.0 for absorption coefficients lower than

15 Mm$^{-1}$ at 450 nm (Rizzo et al., 2011). For BB aerosol particles, the $\mathring{a}_{abs}$ is usually higher than 1.0. However, it depends on the burning conditions, its BC to OA ratio (Saleh et al., 2014), and the BC-BrC size distributions and morphologies (Womack et al., ref needed). Several studies have used the absorption spectral dependence to apportion the fossil fuel and BB contributions to total absorption (Favez et al., 2010; Massabò et al., 2015; Sandradewi et al., 2008). However, the $\mathring{a}_{abs}$ values do not



always reflect the combustion type and using it as a source apportionment parameter could lead to erroneous results (Garg et al., 2016; Lack and Langridge, 2013; Lewis et al., 2008; Wang et al., 2016b). Several measurement studies assume a BC $å_{abs}$ of 1.0 but models show that pure BC could exhibit a broader range of $å_{abs}$ values (Moosmüller et al., 2011). In order to retrieve the ambient BC wavelength dependence, Wang et al. (2016b) proposed the use of the wavelength dependence of $å_{abs}$ instead of $å_{abs}$

itself. The so-called *wavelength dependence of $å_{abs}$* (WDA) is calculated as the difference of two wavelength pairs; one for shorter to long wavelengths (e.g., 440 – 870 nm) and another for medium to long wavelengths (e.g., 675 – 880 nm).

Precise BC mass measurements are required to retrieve the correct relationship between absorptivity and BC mass, defined as the mass absorption cross-section (MAC or $\alpha_{abs}$). The BC mass concentration

has been traditionally measured by using thermal or thermal-optical techniques. However, these methods suffer from several bias, like organic carbon charring that increases the apparent BC concentration, especially when high organic fractions are present (Andreae and Gelencsér, 2006). More recently, laser-induced incandescence (LII) techniques have been introduced (Snelling et al., 2005). These techniques measure the volume-equivalent mass of refractory black carbon (rBC) that vaporizes

at temperatures of 2800-4000 K. The MAC is used by atmospheric radiative transfer models to obtain absorption coefficients from mass concentration data. The MAC of BC varies between 4 and 11 $m^2\ g^{-1}$ at 550 nm (Bond and Bergstrom, 2006), having an average of 6.5 $m^2\ g^{-1}$ at 637 nm for fresh soot. In case of condensation of non-BC material on the BC particles, the MAC can be enhanced due to the well-known 'lensing effect' (Fuller et al., 1999). This commonly happens when BC is emitted by BB,

since it is co-emitted with large amounts of organic vapors that would condense on BC particles (Saleh et al., 2014). Black carbon particles can also obtain a secondary organic aerosol (SOA) coating during advection over the rain forest (Pöschl et al., 2010) as well as inorganic coatings, which has been previously observed at the ATTO site (Pöhlker et al., 2014). It has been found that the coating mass significantly affects the absorption enhancement of BC cores but no significant changes are caused by a

different coating's O:C ratio (Tasoglou et al., 2017). A wide range of MAC can be found in the literature for different fire conditions (smoldering and flaming).



Commonly, the absorption properties of an aerosol population are reported as the single scattering albedo (SSA, $\omega_0$), which is defined as total scattering divided by total extinction (absorption + scattering). Therefore, a lower $\omega_0$ is associated with a stronger absorption. Tropical Amazonian forest

fires have moderately high $\omega_0$ values (0.93 ± 0.02 at 670 nm), given the high amount of scattering aerosols which are co-emitted with LAC, compared to African savanna fires that have lower $\omega_0$ values (0.84 ± 0.015 at 670 nm)  (Reid et al., 2005). In the Amazon rain forest, long-term measurements by Rizzo et al. (2013) have found similar values for $\omega_0$ during the dry and the wet season, 0.87 ± 0.06 and 0.86 ± 0.09, respectively. The low $\omega_0$ in the wet season is attributed to long-range transport aerosol

masses that include mineral dust and aged BB aerosol particles. Aged BB aerosol is proven to have increased MAC, and therefore lower $\omega_0$ (Reid et al., 2005). Moreover, part of the biogenic aerosol can contribute up to 35 % of total light absorption (Guyon et al., 2004).

When present in large mass amounts in the atmosphere, mineral dust can significantly absorb light, with a MAC of 0.02 – 0.1 m² g⁻¹ at 550 nm (Clarke and Charlson, 1985). It is mobilized from soils and

suspended in the atmosphere by windstorms in areas like the Saharan desert in Africa. Dust aerosol particles in the atmosphere efficiently scatter visible radiation and are able to absorb infrared radiation (Andreae, 1996), having a $\mathring{a}_{abs}$ >> 1.0 (Caponi et al., 2017; Denjean et al., 2016). Mineral dust plumes travel over the Atlantic Ocean and are able to reach the American continent. Depending on the circulation patterns over the tropical Atlantic, the African dust plumes will be transported to South

America or to the Caribbean Sea and Central America (Prospero et al., 1981). The average transport time from emission to deposition in the Amazon basin during winter is ~10 days (Gläser et al., 2015). Ground measurements of aerosol physical and chemical properties have confirmed that between January and April mineral dust plumes from Africa episodically dominate the aerosol load over the Amazon rain forest (Formenti et al., 2001; Guyon et al., 2004; Moran-Zuloaga et al., 2017; Talbot et al.,

1990; Wang et al., 2016a). Moreover, the dust-enriched aerosol usually arrives together with BB aerosol emitted by fires in the sub-Sahelian west Africa and also aerosol particles emitted by industrial activities in Morocco and the western Sahara coast (Pöhlker et al., 2017a; Salvador et al., 2016). In spite of anthropogenic disturbance of soils in Africa that could enhance the flux of mineral dust to the



atmosphere (Andreae, 1991), a decreasing trend in mineral dust emissions since the 1980s has been observed and is mainly caused by a reduction of surface winds in the Sahel region (Ridley et al., 2014).

This study provides a comprehensive and in-depth analysis of the aerosol optical properties in the Amazonian atmosphere. A continuous long-term dataset (2012 – 2017) of different optical properties is provided. We especially focus on the impact of BB emissions from long-range transport and from regional/local open fires during the dry season. By using data from another central Amazonia remote

sampling site, we extend our time series back to 2008 and provide the longest optical properties dataset measured in the Amazon rain forest. By this means, we are able to study the perturbations caused by El Niño Southern Oscillation (ENSO), which has been reported to cause drought all over the Amazon Basin (see Fig. S1), with increasing fire activity and forest degradation (Aragão et al., 2008; Lewis et al., 2011).

**2 Materials and methods**

**2.1 Sampling site and measurement period**

Aerosol particles and trace gases are being measured at the Amazon Tall Tower Observatory (ATTO) site, located in the Uatumã Sustainable Development Reserve, Amazonas State, Brazil, in central Amazonia since 2012 (Andreae et al., 2015). The large-scale meteorological conditions of the site are

determined by the seasonal shifts of the Inter-Tropical Convergence Zone (ITCZ) location (Pöhlker et al., 2017a). From August to November, during the *dry season*, the ITCZ is located in the north of South America, and mostly Southern Hemisphere air masses reach the ATTO site bringing BB emissions from deforestation hotspots in Southeastern Brazil (i.e., so called arc of deforestation) as well as transcontinental emissions from Southern Africa. During the *wet season*, from February to May, when

the ITCZ shifts to southern latitudes, the air masses generally come from the northern hemisphere, following a path over the Atlantic Ocean from the African continent and then, over mostly untouched forest areas upwind of the ATTO site. The transition seasons, *dry to wet* and *wet to dry*, occur in December – January and June – July, respectively.





At the ATTO site, aerosol measurements were started in March 2012, being continuously extended and
intensified since then. In the course of this process, the aerosol inlet system was modified and upgraded
stepwise. A detailed list of the different inlet configurations and locations can be found in Table S1. On
04 May 2014, a $PM_1$ cyclone was installed in the common inlet line for the aerosol optical
measurements. The rest of the instrumentation kept sampling total suspended particles (TSP). The
sampled aerosol was dried by diffusion driers filled with silica gel to guarantee a relative humidity
around 40 % or below. An automatic regenerating adsorption aerosol dryer (Tuch et al., 2009) was
installed in January 2015.

Another sampling site, ZF2 / TT34 tower, located 60 km NNW of Manaus (Fig. S2) has been the
location of long-term aerosol observations and intensive measurement campaigns (Rizzo et al., 2013).
Given that most of the air masses that reach the ZF2 site are the same that are transported over the
ATTO site (Pöhlker et al., 2017a), the ZF2 data is usually comparable to the ATTO data and the time
series presented in this study can complement previous ZF2 time series already reported for the period
2008 – 2011 (Rizzo et al., 2013). Additionally, two intensive observation periods (IOP) and long-term
measurements of the GoAmazon2014/5 experiment took place at several measurement sites in the
Amazon Basin, including the ATTO site. More details can be found in Martin et al. (2016, 2017).

**2.2 Instrumentation**

**2.2.1 Aerosol light scattering measurements**

Scattering coefficients at ATTO were measured using different nephelometers. Figure S3 shows the
measurement periods of the different instruments. The first one was a 3-wavelength integrating
nephelometer (Model 3563, TSI, St. Paul, USA) (14 Aug 2012 to 24 Nov 2013). The instrument
measures aerosol scattering ($\sigma_{sp}$) and backscattering ($\sigma_{bsp}$) at 450, 550 and 700 nm (Anderson et al.,
1996). Calibrations were periodically done by using $CO_2$ as span gas. Given the optical configuration of
the instrument, the truncation of forward scattered radiation constitutes the largest source of error and
was corrected by following the procedure described by Anderson et al. (1996). The estimated error of
the nephelometer measurements is 8 % for scattering coefficients in the order of 10 Mm$^{-1}$ (Rizzo et al.,



2013). Using an averaging time of 30 min, the detection limit at 550 nm was 0.14 Mm$^{-1}$ (Rizzo et al., 2013).

Later, in February 2014, the TSI nephelometer was replaced by an Aurora 3000 (Ecotech Pty Ltd., Knoxfield, Australia), which measures at 450, 525, and 635 nm wavelength. Over the measurement period studied in this work, we used two different Aurora instruments, with and without backscattering.

The instrument was set up to work with an integration time of 1 min. Similar to the TSI nephelometer, $CO_2$ calibrations were periodically performed. Uncertainty in scattering measurements by the Aurora nephelometers was estimated to be 5 % (Müller et al., 2011).

### 2.2.2 Aerosol light attenuation and absorption measurements

Light absorption coefficients at 637 nm wavelength, $\sigma_{ap\ 637}$, were measured by a multi-angle absorption

photometer, (MAAP, model 5012, Thermo Electron Group, Waltham, USA). This instrument measures the transmission of light through a glass-fiber filter on which aerosol particles are collected. Additionally to the forward hemisphere transmission measurement, the MAAP measures the light back scattering at 130° and 165°. By using a radiative transfer model (Petzold and Schönlinner, 2004), the instrument is able to provide absorption coefficients with a time resolution of 5 min. The provided data

are 1-min running averages. By averaging the data at 30-min intervals, the MAAP detection limit is 0.132 Mm$^{-1}$, which corresponds to a $BC_e$ mass concentration of 20 ng m$^{-3}$ (calculated with a MAC of 6.6 m² g$^{-1}$). The MAAP was generally operated at a flow rate of 10 L min$^{-1}$, but for some periods the flow rate was reduced to 8.3 L min$^{-1}$. According to Müller et al. (2011), the MAAP measures at a wavelength of 637 ± 1 nm, instead of the 670 nm reported in the instrument's manual. In our

calculations, we use 637 nm as the default MAAP wavelength and do not apply any interpolation factor to scale up the data from 670 to 637 nm since it would be in the ~5 % range, which is within the instrument uncertainty. The total uncertainty of the MAAP absorption measurements is of the order of 10 % for 30-min average times (Rizzo et al., 2013).

An Aethalometer was used to measure attenuation of light by aerosol particles at different wavelengths.

This instrument uses a LED light source to irradiate an aerosol-laden quartz-fiber filter and a detector,




located in the forward hemisphere, to measure the light transmission (Hansen et al., 1984). The measured transmission is compared to a blank measurement in order to obtain a change in light transmission (attenuation). This attenuation is then converted to BC mass concentration by using a mass attenuation cross section that depends on the instrument model (14625 and 6837.6 $m^2\ g^{-1}\ \lambda^{-1}$ for the

AE31 and AE33 Aethalometer models, respectively).

Aethalometer measurements started at the ATTO site in April 2012 using an Aethalometer model AE31 (Magee Scientific, Berkeley, USA). The instrument was operated at different flow rates during the measurement period (varying from 2.0 to 3.7 L $min^{-1}$) and measured attenuation every 15 min. In January 2015, a new Aethalometer, model AE33 (Aerosol d.o.o., Ljubljana, Slovenia), was installed.

The overlapping measurement time of the AE31 and the AE33 models (27 Nov to 15 Dec 2014) enabled the comparison of both datasets. We found a good agreement between both models (difference < 10 %) for measurements at 470, 520, 590, and 660 nm. However, the wavelength dependence did not fit very well during this intercomparison period. Similar deviations in the wavelength dependence of AE31 and AE33 have been reported previously (ACTRIS, 2014). Nevertheless, it is still not clear if the

higher wavelength dependence of the AE33 compared to the AE31 is the result of an artifact of the instrument. An independent multi-wavelength absorption measurement can help to clarify the aforementioned AE31/AE33 deviation in $\mathring{a}_{abs}$ (Saturno et al., 2017b). The comparison between compensated AE31 and AE33 data was used to correct the AE33 wavelength dependence deviation by applying intercomparison factors to AE33 data. The obtained AE31-AE33 intercomparison fits are

shown in Fig. S4.

Aethalometer data require several corrections to account for different artifacts related to multiple scattering by the filter fibers, scattering by embedded aerosol particles and filter loading effects. The correction applied in this study has been described in a previous article (Saturno et al., 2016). The compensation algorithm is based on the correction scheme proposed by Collaud Coen et al. (2010). It

uses MAAP data as a reference absorption measurement, which could introduce uncertainties related to the modification that aerosol particles can suffer by being deposited on a filter matrix. We retrieved the $\mathring{a}_{abs}$ from applying a log-log fit to Aethalometer data corrected for filter-loading and multiple scattering effects. In the case of the Aethalometer AE33, the measurements do not require a filter-loading



correction because this model uses the dual-spot technology which accounts for this artifact (Drinovec
et al., 2015).

### 2.2.3 rBC mass measurements and MAC calculations

Refractory black carbon (rBC) was measured using a single particle soot photometer (SP2) revision C
(Droplet Measurement Technologies, Longmont, USA). Initially, the measurements were done with a
4-channel SP2 and the instrument was upgraded on 19 January 2015 to the 8-channel configuration.
Figure S3 shows the different measurement periods of this instrument. The SP2 uses a high-intensity
Nd:YAG laser beam (1 MW cm$^{-2}$, $\lambda = 1064$ nm) to irradiate aerosol particles that are provided by an air
jet at 90°, with a flow rate of 0.12 L min$^{-1}$. All particles scatter the light from the laser beam and some
of them, which are able to absorb radiation at the given wavelength (e.g., rBC), will incandesce and
vaporize at high temperatures (Moteki and Kondo, 2008; Stephens et al., 2003). Four avalanche
photo-diode (APD) detectors are installed in the instrument to measure a) scattering, b) broadband
incandescence (350 – 800 nm), c) narrowband incandescence (630 – 880 nm) and d) scattering with a
split detector. Time dependent data is recorded from each particle as it passes through the laser beam.
The ratio between broadband and narrowband signals can provide information on the particle's
composition since it is related to the boiling point temperature of the sampled particles (Schwarz et al.,
2006). The instrument was periodically calibrated using fullerene soot (Alfa Aesar Inc.) as rBC
reference material. A quadratic fit was applied to the recorded incandescence peak heights vs. the mass
of mobility size-selected fullerene particles. The fullerene effective densities were taken from Gysel et
al. (2011). The scattering detector was calibrated using polystyrene latex spheres (PSL) by relating the
scattering signal to the PSL scattering cross-section. The SP2 rBC dynamic ranges were 80 – 280 nm
and 80 – 450 nm for the 4-channel and the 8-channel configurations, respectively.

The narrow dynamic range of the 4-channel SP2 was preventing us from measuring rBC mass
concentration values comparable to MAAP measurements. In a comparison with another 8-channel
instrument during the GoAmazon2014/5 experiment we found that the 4-channel instrument was
underestimating the rBC mass concentration by a factor of 40 %. This factor was stable during the wet





season 2014 but we could not guarantee or measure its stability during the dry season. Due to instability

of this factor over the sampling period, a proper data correction was not possible. Therefore, in this

paper we use only the 8-channel instrument's data, which were available from 09 February 2015 until 31

July 2016 with some interruptions due to hardware failures. The 8-channel SP2 rBC mass measurement

was underestimated by a factor of 5 %, related to the size-dependent detection efficiency of the

instrument, which is below 100 % in the 80 to 150 nm diameter range. Therefore, a scaling factor of

1.05 was applied to rBC mass concentration data to account for this systematic error.

The BC mass absorption cross-section, $\alpha_{abs}$, was calculated by running daily fits of 30-min averaged

MAAP $\sigma_{ap\,637}$ vs. SP2 rBC mass concentration data, using a standardized major axis estimation (as

explained in section 2.6). Fits with $R^2 < 0.9$ were filtered out resulting in a total of 106 out of 220 days

included in the final result. The obtained $\alpha_{abs}$ values (shown in section 3.1) were used to convert MAAP

absorption measurements into $BC_e$ mass concentrations.

### 2.2.4 Complementary measurements

Online chemical composition of aerosol particles has been measured since August 2014 using an

aerosol chemical speciation monitor (ACSM) (Aerodyne Research Inc., Billerica, USA). Initial results

on non-refractory aerosol chemical composition at the ATTO site have already been reported by

Andreae et al. (2015) and a detailed paper on the long-term ACSM observations is being prepared by

Carbone et al. (2017). This online mass spectrometry technique detects organics, nitrate, sulfate,

ammonium and chloride in the sub-micron (< 1 μm) aerosol size range (Ng et al., 2011).

A Picarro cavity ring-down spectrometer G1302 analyzer (Picarro Inc., Santa Clara, USA) measured

$CO_2$ and CO at the ATTO site. Three calibration tanks were used to calibrate the instrument every

100 h. A Nafion dryer was installed in front of the instrument in order to reduce the noise in the CO

measurements, which are affected by the high relative humidity of the tropical forest air. Calibration

and performance checks will be reported in an upcoming paper. The instrument samples at five different

heights but we restrict our analysis to the data measured at 79 m. All CO measurements have been

conducted on the walk-up tower. More details on the measurement setup can be found in Winderlich et



al. (2010). In order to calculate the BC enhancement ratios with respect to CO ($ER_{BC}$), we used a major axis estimation fit that was applied to the bivariate data (Falster et al., 2006) where the slope represents the enhancement ratio. The 5th percentiles were used as background values.

Condensation nuclei (CN) number concentrations, $N_{CN}$, and size distributions from 10 nm to 10 μm

were continuously measured using several instruments including mobility and optical particle sizers (more details can be found in Andreae et al. (2015)). In this study, we used coarse mode (> 1 μm) number and mass concentrations obtained by means of an optical particle sizer (OPS) model 3330 (TSI Inc., Shoreview, USA) to identify mineral dust transport events. A detailed analysis of the Saharan dust plumes arrivals at the ATTO site can be found in Moran-Zuloaga et al. (2017). Aerosol particle size

distributions (10 – 430 nm diameter) were measured with a scanning mobility particle sizer (SMPS) models 3080 and 3081 (TSI Inc., Shoreview, USA) using a condensation particle counter (CPC), model 3772 (TSI Inc., Shoreview, USA).

## 2.3 Wavelength dependence calculations

Light scattering and absorption wavelength dependence are represented by the Ångström exponents, $\mathring{a}_{sca}$

and $\mathring{a}_{abs}$, respectively. The Ångström exponent can be retrieved when measurements at two different wavelengths are available, for example, the $\mathring{a}_{abs}$ can be calculated as

$$\mathring{a}_{abs} = -\frac{\ln\left(\frac{\sigma_{ap}(\lambda_1)}{\sigma_{ap}(\lambda_2)}\right)}{\ln\left(\frac{\lambda_1}{\lambda_2}\right)} \quad , \tag{1}$$

where $\sigma_{ap}$ is the absorption coefficient at two different wavelengths, $\lambda_1$ and $\lambda_2$.

When measurements at more than two wavelengths are available, a linear fit can be used to retrieve the

Ångström exponent from the logarithm of the absorption (or scattering) coefficients vs. the logarithm of the wavelength, as follows

$$\ln \sigma_{ap} = -\mathring{a}_{abs} \ln(\lambda) + \ln(\text{constant}) \quad , \tag{2}$$





Black carbon is commonly taken to be wavelength-independent with $\mathring{a}_{abs} = 1$. However, this assumption is theoretically wrong and the BC-related $\mathring{a}_{abs}$ is very sensitive to the size of the particles (Moosmüller et al., 2011). Wang et al. (2016b) proposed a method to calculate the *wavelength dependence of the Ångström exponent* (WDA) in order to estimate the BrC contribution to total light absorption by aerosol particles. They use the difference between two $\mathring{a}_{abs}$ calculated for two different wavelength pairs (440 – 870 nm, and 675 – 880 nm) using aerosol robotic network (AERONET) and Aethalometer data. We use a similar approach to retrieve WDA using Aethalometer data from the ATTO site. In this study the WDA is calculated as follows:

$$WDA = \mathring{a}_{abs\ 370\text{-}950} - \mathring{a}_{abs\ 660\text{-}950} \quad , \tag{3}$$

where $\mathring{a}_{abs\ 370\text{-}950}$ and $\mathring{a}_{abs\ 660\text{-}950}$ correspond to the absorption Ångström exponents calculated for the 370 – 950 and 660 – 950 nm wavelength pairs, respectively.

Theoretical WDA values were calculated following conceptual Mie theory models for (i) polydisperse BC particles (Mishchenko et al., 1999), and (ii) core-shell internally mixed monodisperse BC (Bohren and Huffman, 1983). Calculated BC WDA thresholds, presented in Fig. S5, were compared to the ambient data in order to retrieve the BrC contribution to light absorption. Characteristic BC core size distributions measured by the SP2 during the wet and dry season were used in the polydisperse BC-only model to retrieve extinction efficiency and single scattering albedo. The refractive indices used were 1.95 - 0.79i for BC (Bond and Bergstrom, 2006) and 1.55 - 0.001i for the coating material (Liu et al., 2015). The latter value was only used for the internally mixed BC case. The BC core diameters used in the internally mixed case were 100, 125, 150, 175, 200, 225, and 250 nm, with coating thickness to core size ratio from 0.1 to 1. Black carbon density was set to 1.8 g cm$^{-3}$ (Schkolnik et al., 2007). Brown carbon absorption at 370 nm was calculated by using the WDA approach, as follows:

$$BC\,\mathring{a}_{abs\ 370\text{-}950} = \mathring{a}_{abs\ 660\text{-}950} + WDA \quad , \tag{4}$$

$$BC\,\sigma_{ap\ 370} = \sigma_{ap\ 950} \times \left( \frac{370}{950} \right)^{-BC\,\mathring{a}_{abs\ 370\text{-}950}} \quad , \tag{5}$$

$$BrC\,\sigma_{ap\ 370} = \sigma_{ap\ 370} - BC\,\sigma_{ap\ 370} \quad . \tag{6}$$



The uncertainties of the BrC contribution to total absorption at 370 nm were calculated using the theoretical minimum and maximum WDA values. They were below 37 % overall, and decreased to

below 19 % when the BrC contribution was higher than 30 % at 370 nm. The relative overestimation of the BrC contribution obtained by using different BC core sizes and different refractive indices in the Mie model calculations can be found in Table S2.

**2.4 HYSPLIT backward trajectories and clustering**

The systematic back trajectory analysis used here is described in Pöhlker et al. (2017a). Briefly

summarized: Three-days backward trajectories were calculated by running the NOAA hybrid single-particle Lagrangian integrated trajectory (HYSPLIT) model (Draxler and Hess, 1998) using 1-degree resolution meteorological data from the global data assimilation system (GDAS1). The trajectories were calculated for 1000 m above ground level at 1 hour intervals for the period January 2008 to June 2016. The entire trajectory ensemble was classified into 15 backward trajectory (BT)

clusters using a k-means cluster analysis. The clusters represent different air mass transport tracks and velocities. The different cluster average trajectories and their frequency of occurrence are shown in Fig. 1a and 1b, respectively. The clusters are classified as north-easterly ("NE1", "NE2", and "NE3"), east-north-easterly ("ENE1", "ENE2", "ENE3", and "ENE4"), easterly ("E1", "E2", "E3", and "E4), south-easterly ("ESE1", "ESE2", and "ESE3"), and south-westerly ("SW1") trajectory clusters.

South American fire count data were retrieved from the satellite observations database available online by the Instituto Nacional de Pesquisas Espaciais (INPE), Brazil, at https://prodwww-queimadas.dgi.inpe.br/bdqueimadas/, last access on 04 Apr 2017. The fire data covered the same period as the HYSPLIT clustering analysis period, January 2008 to June 2016. Fire counts were classified according to the corresponding BT cluster where they occurred at hourly resolution. The fire counts

reported in this study were weighted according to the trajectory density as (trajectory counts) / 100 km$^2$. Since the fire count number depends on the amount of satellite data available, we use these data with caution and only as a qualitative reference. For an extended discussion on fire geographical locations and land cover types, see Pöhlker et al. (2017).



## 2.5 Satellite data

The aerosol optical depth (AOD) at 550 nm, measured by the moderate resolution imaging spectroradiometers (MODIS) on board of the satellites Terra and Aqua, was retrieved for two domains of interest (see Fig. 2a):

- DOI1: Over the Atlantic Ocean. Used to monitor the westward transport of BB aerosol particles from southern Africa, which is mostly emitted during the Amazon dry season, especially

between August and September (Das et al., 2017). There is no guarantee that the observed aerosol over this area will necessarily reach the ATTO site, but it is used as an indication of LRT events from southern Africa that will likely reach the Amazon Basin.
   Boundaries: 30 W; 20 S; 10 W; 0 S.

- DOI2: Over the southern Amazon. Used to monitor BB in this region where fire activity is

related to deforestation and agriculture-related activities.
   Boundaries: 58 W; 14 S; 40 W; 8 S.

The MODIS products can be found online on the Goddard Earth Science Data and Information Services Center at https://giovanni.gsfc.nasa.gov/giovanni/, last access on 17 Jul 2017, (GES-DISC, 2017).

Terra and Aqua data were averaged over the two different domains. The averaged AOD at 550 nm time

series corresponding to DOI1 and DOI2 can be found in Fig. 2b. The seasonality observed for both datasets is similar but AOD for DOI1 (Atlantic Ocean) generally increased in August and decreased after the end of September with some peaks in January – February, especially in 2016. On the other hand, high AOD values in DOI2 (South Amazon), increased sharply in the beginning of September and decreased continuously until the middle of December with the exception of the dry season 2015 when

high AOD was observed until February 2016.

## 2.6 Data treatment

The analyzed data were averaged to 30-min intervals and corrected to standard temperature and pressure (STP, 273.15 K and 1013.25 hPa). Furthermore, the scattering data were interpolated to 637 nm to compare directly to the absorption data obtained by the MAAP, in order to avoid the





uncertainty associated with the absorption spectral dependence calculation. The time periods of major

and medium dust influence were taken from a study by Moran-Zuloaga et al. (2017). During the dry

season, BB pulses were segregated by using the 75$^{th}$ percentile of $\sigma_{ap\,637}$ as a threshold. When examining

correlations between independent measurements, we applied standardized major-axis estimations

(SMA) by using the SMATR package (Falster et al., 2006) in the statistical software environment R

(R Development Core Team, 2009). This method minimizes the error on the $x$ and $y$ axes and not only

at the $y$ axis, like a linear regression does. Therefore, it provides unbiased estimates of the slope

(Warton et al., 2006).

## 3 Results and discussion

### 3.1 Overview of aerosol optical properties (2012 – 2017)

This section summarizes the aerosol optical properties from five years of continuous measurements at

the ATTO site. The corresponding time series are shown in Fig. 3 and descriptive statistics can be found

in Table 2. The wet and dry season statistics were calculated excluding the transition periods.

The scattering coefficients, $\sigma_{sp}$, shown in Fig. 3a, averaged $7.5 \pm 9.3$ Mm$^{-1}$ and $33 \pm 25$ Mm$^{-1}$ at 550 nm

during the wet and the dry season, respectively (see Table 2). These values agree well with previously

reported results at ZF2 of $8.1 \pm 7.2$ Mm$^{-1}$ and $36 \pm 48$ Mm$^{-1}$ at 550 nm during the wet and dry season,

respectively (Rizzo et al., 2013). The same is valid for our results at 450 nm and 700 nm (not shown)

and the ones presented by Rizzo et al. (2003). The proximity of both sites, ATTO and ZF2, frequently

allows probing comparable air masses of similar origin and atmospheric history. The long-term

measurements show also a pronounced year-to-year variability in $\sigma_{sp}$ (compare e.g., 2014 and 2015).

The largest observed deviations from the dry-season average were found during the dry season 2015

with an average increase of 38 % in $\sigma_{sp}$ at 550 nm. Similar increases were observed in $\sigma_{sp}$ at 450 and

637 nm. These increases can be directly related to the higher occurrence of fire episodes during the

strong ENSO period 2015/6 with its negative precipitation anomaly, as discussed in more detail in

sections 3.5 and 3.6.



The absorption coefficients, $\sigma_{ap}$, at 637 nm (MAAP) are shown in Fig. 3b, and averaged $0.68 \pm 0.91$

Mm$^{-1}$ and $4.0 \pm 2.2$ Mm$^{-1}$ during the wet and the dry season, respectively. Also for this parameter,

comparable values were measured at the ZF2 site, with averages of $1.0 \pm 1.4$ Mm$^{-1}$ and $3.9 \pm 3.6$ Mm$^{-1}$

at 637 nm during the wet and the dry season, respectively (Rizzo et al., 2013). The higher increase of

the absorption coefficient (factor of 5.9) from wet to dry season compared to the increase in scattering

(factor of 4.4) affected the $\omega_0$ (see Fig. 3c). Lower values were observed during the dry season

($0.87 \pm 0.03$ at 637 nm, $0.81 \pm 0.08$ at 550 nm) compared to the averages observed in the wet season

($0.93 \pm 0.04$ at 637 nm, $0.88 \pm 0.08$ at 550 nm). At the ZF2 site, Rizzo et al. (2013) have found small

differences between $\omega_0$ values during the dry and wet seasons ($0.87 \pm 0.06$, and $0.86 \pm 0.09$ at 637 nm,

respectively) for over 2 years (2008 – 2011) measurements. However, measurements during the wet

season in 1998 at a sampling site closer to the ATTO site (Balbina, 60 km NW of ATTO and 140 km

NE of Manaus) showed higher $\omega_0$ values: $0.92 – 0.95$ at 550 nm (Formenti et al., 2001). These values

are within our measurement range for the same season ($0.88 \pm 0.08$ at 550 nm). Single scattering albedo

retrieved from multi-year ground-based radiometer measurements in the Amazonian forest had an

average of $0.93 \pm 0.02$ (Dubovik et al., 2002). Given that we sampled dried aerosol particles, our

average $\omega_0$ are expected to be lower than these ambient-humidity values during the entire measurement

period and the dry season. Measurements close to BB sources in Brazil have shown a wide range of $\omega_0$;

e.g., Chand et al. (2006) found a $\omega_0$ of $0.92 \pm 0.02$ (550 nm) for dried aerosol over Rondônia, whereas

Guyon et al. (2003) calculated lower $\omega_0$ values during BB events at the end of the LBA-EUSTACH 1

campaign in Rondônia, reaching $0.85 \pm 0.02$ at 550 nm. Fresh smoke fires have been reported to have

lower $\omega_0$, of $0.79 \pm 0.05$ at 550 nm (Reid et al., 1998).

The scattering Ångström exponent, $\mathring{a}_{sca}$, is a function of the aerosol particle size distribution. Rizzo et al.

(2013), however, pointed out that this relationship is only evident for surface and volume mean

diameters and was not clearly valid between $\mathring{a}_{sca}$ and count mean diameters. We obtained higher $\mathring{a}_{sca}$

values during the dry season ($1.71 \pm 0.24$) compared to the wet season ($1.29 \pm 0.50$) as shown in

Fig. 3d. This is an indication of the dominance of fine mode aerosol (mostly BB related) during the dry

season over the coarse mode aerosols that become more important in the wet season (i.e., PBAP,

Saharan dust and sea salt), as previously observed at the ATTO site (Andreae et al., 2015;

35                                                                 18



Moran-Zuloaga et al., 2017). A similar seasonal trend has been observed at the ZF2 site, where $\mathring{a}_{sca}$ was

1.70 ± 1.41 and 1.48 ± 1.12 (30-min averages) for the dry and the wet season, respectively (Rizzo et al.,

2013). A detailed analysis of the coarse mode aerosol abundance and properties measured at the ATTO

site is presented elsewhere (Moran-Zuloaga et al., 2017).

Regarding the absorption Ångström exponent, $\mathring{a}_{abs}$, the overall average during the whole sampling

period was 0.93 ± 0.16 (see Fig. 3e). Although no significant difference was found between dry and wet

season averaged values, the $\mathring{a}_{abs}$ was slightly higher during the dry season, reaching an average of

0.94 ± 0.16 compared to a wet season average of 0.91 ± 0.19. The Aethalometer compensation

calculation could potentially affect the retrieved $\mathring{a}_{abs}$ values. It has been shown that the raw attenuation

Ångström exponent can represent a good approximation to the real $\mathring{a}_{abs}$ (Saturno et al., 2017b). High

absorption and scattering coefficients coincide with ESE and E trajectories, which are mostly dominant,

but not exclusively, during the dry season, see Fig. 1. On the other hand, the "cleanest" periods in the

wet season, when light absorption reaches its minimum and $\omega_0$ its maximum, the dominant trajectories

are ENE and NE.

Accurate MAC values are required to retrieve BC mass concentrations from absorption measurements.

During the entire measurement period, the calculated MAC was 11.9 ± 1.4 m² g⁻¹ (mean ± standard

deviation) at λ = 637 nm. Daily calculated MAC values in the wet season were slightly lower on

average compared to the dry season values (11.4 ± 1.2 and 12.3 ± 1.3 m² g⁻¹, respectively, see Table 2).

As an illustration of the different MAC values obtained in the wet and the dry season, $\sigma_{ap\,637}$ vs. $M_{rBC}$

scatter plots are presented as supplementary information in Fig. S6. Lower MAC values measured in the

wet season 2016 could be associated with less coated BC compared to more aged particles in the dry

season, which could have thicker coatings. Nevertheless, both values are much higher than the

6.6 m² g⁻¹ suggested by Bond and Bergstrom (2006), especially considering that mineral dust and BrC

do not strongly absorb at this wavelength and would therefore have little influence on the apparent

MAC. However, they are in agreement with a modelled absorption enhancement of 1.6 calculated for

open biomass burning in Brazil (Liu et al., 2017). In any case, there are large discrepancies that make it

difficult to compare different MAC values obtained from ambient measurements due to systematic

analytical uncertainties that dominate over the natural variability (Zanatta et al., 2016). These



uncertainties are introduced by filter-based absorption measurement biases and BC mass over- or underestimation when thermal optical methods are used. In the case of the SP2, the rBC mass measurements are free of the different biases that affect thermal-optical techniques and are a wavelength independent measurement. In the case of absorption measurements, a positive bias is

introduced when organic aerosol deposits on the filter, enhancing the scattering by the filter fibers and the absorption by previously deposited BC particles when coating them. This artifact can be between 12 and 70 % for particle soot absorption photometer (PSAP) measurements and will depend on the OA to BC ratio and the aging state of the organic aerosol particles (Lack et al., 2008). We expect a lower artifact for the MAAP since the scattering by filter fibers is accounted by the reflectance measurements,

but using our instrumentation we are not able to estimate the artifact coming from embedded BC absorption being modified by organic aerosol deposition. There are only few field studies that present comparisons of rBC measurements and light absorption measurements, like MAAP, photoacoustic spectrometry (PAS), or Aethalometer, and especially long-term measurements are scarce. Raatikainen et al. (2015) reported SP2 (8-channel) and MAAP measurements in the Finnish Arctic and found that

SP2 rBC mass concentrations were 5 times lower than MAAP $BC_e$ mass concentration measurements, which is equivalent to MAC values of ~30 m² g⁻¹ at 637 nm. Some other studies have found values in closer agreement with our ATTO MAC results. For example, Laborde et al. (2013) found that air masses over Paris had an average MAC of 11.9 and 10.8 m² g⁻¹ (interpolated to 637 nm), for aged and fresh BB aerosol, respectively. Additionally, Liu et al. (2010) calculated a median MAC of

10.2 ± 3.2 m² g⁻¹ during a measurement campaign at the Jungfraujoch research station in Switzerland. Another study in Mexico City, using PSAP for absorption measurements at λ = 660 nm, found a MAC of 11.2 m² g⁻¹ (interpolated to 637 nm) (Subramanian et al., 2010).

## 3.2 Variability of optical properties during the dry season

The Amazonian dry season is generally impacted by BB aerosol particles that cause an increase in

scattering and absorption coefficients (see Fig. 3a-b). However, the aerosol optical properties vary with the burning material (and region), as well as the aging process prior to reaching the ATTO site. In order to study the dry season variability of BB aerosol particles, multi-year (2012 – 2017) weekly averages





were analyzed. The air mass trajectories, presented as BT clusters in Fig. 4a, show a decreasing

dominance of ESE winds from August to November, whereas from October to November there is an

increasing influence of ENE winds, indicating the south-to-north air mass trajectory shift that occurs

during the transition from the dry to the wet season. It is important to note that southerly and easterly

winds are most likely to bring BB aerosol to the ATTO site during the dry season, given that very active

open fire areas during this period are located in the southern Amazon and the Cerrado region (Andreae

et al., 2012; Guyon et al., 2005) and, more remotely, in southern Africa (Andreae et al., 1994; Barbosa

et al., 1999; Das et al., 2017). Aerosol optical depth at 550 nm is used in this study as a parameter to

study the seasonal pattern of BB emission transport from both areas. In section 2.5, we defined two

domains of interest to study the aerosol seasonal patterns in these two areas: DOI1 for the LRT of South

African smoke over the Atlantic Ocean, and DOI2 for the fires occurring in the southern Amazon. For

the case of southern Africa fires (DOI1), the seasonal pattern shows an important influence during

August – October, slightly decreasing towards the end of the Amazonian dry season (see Fig. 4d). For

the southern Amazon region (DOI2), the typical fire seasonality during the dry season is observed in the

AOD over this area (Fig. 4d) with the highest values observed during September and October. A second

increase in AOD is observed in the middle of November over DOI2. It is important to note that August

seems to be the period when African LRT is a more important source than regional emissions and could

be considered as the main contributor of BB aerosol to the ATTO site during this time. For the rest of

the dry season, it is likely that the aerosol properties are defined by South American BB emissions. In

fact, the shift in air mass trajectories and variation of BB sources drive the BrC contribution to $\sigma_{ap\ 370}$, as

can be seen in Fig. 4b. The BrC contribution (associated with high $\mathring{a}_{abs}$) is more important at the end of

the dry season and is lower during August, when the aerosol particles likely arrive from Zambian

woodland savanna fires (Barbosa et al., 1999), which burn more efficiently and emit aerosol particles

with lower $\omega_0$, 0.84 ± 0.015 at 670 nm in average (Dubovik et al., 2002). Additionally, on average, high

$\sigma_{ap\ 637}$ events (see increasing circle size in Fig. 4b) are more likely to bring high BrC containing aerosol,

which is another indication that closer fires have higher probability to provide BrC-rich aerosol particles

to the ATTO site. The absorption wavelength dependence and BrC contribution are discussed in detail

in section 3.6. The differences between both identified BB sources in terms of BrC can be explained by





two reasons: (i) the BrC photochemical oxidation during transport that would strongly affect LRT aerosol, and (ii) a lower wet scavenging rate for BC during transport, which would lead to an increased BC fraction in the aerosol population. In terms of the single scattering albedo ($\omega_0$, Fig. 4c), its increase towards the end of the dry season confirms that the aerosol particles during this time are scattering more

radiation, not only due to higher BrC presence but also due to an increased sulfate concentration.

### 3.3 Diel cycles

Figure 5 presents the different diel cycles observed during the dry and the wet season for selected aerosol properties and some meteorological parameters ($N_{acc}$, $\sigma_{ap\ 637}$, $\sigma_{ap\ BrC\ 370}$, $P_{ATTO}$, and $\theta_e$). In order to study the typical diel cycles in each season, extreme events like mineral dust transport in the wet season

and nearby BB during El Niño in 2015 – 2016 have been excluded. The diel cycle of the equivalent potential temperature (Fig. 6i-j) reflects the evolution of the planetary boundary layer. Shortly before sunrise (~ 10:00 UTC), $\theta_e$ exhibits its minimum and increases afterwards reaching its maximum values in the early afternoon hours. The pronounced temperature increase in the early morning hours is connected to the initiation of vertical mixing, leading to the evolution of the convective boundary layer.

After sunset, a stable nocturnal boundary layer is formed close to the forest canopy. A detailed analysis of the planetary boundary layer of the Amazon can be found in Fisch et al. (2004). Figures 5a-b (dry and the wet season, respectively) show diel cycles of accumulation mode (particle diameter between 100 – 430 nm) particle number concentration, $N_{acc}$. The diel patterns are similar during both seasons, with a minimum at sunrise, and an increase that starts in the morning at 12:00 UTC (8:00 LT) and

maximum concentrations between 17:00 and 18:00 UTC (13:00 – 14:00 LT). This diel pattern observed in $N_{acc}$ is driven by the diurnal evolution of the planetary boundary layer. On the one hand, the stable nocturnal layer leads to a concentration of particles and gases close to the canopy. On the other hand, the canopy acts as an effective particle sink, resulting in a concentration decrease towards the early morning (Ahlm et al., 2009). After sunrise, vertical mixing breaks up the stable nocturnal boundary

layer. While the subsequent increase in $N_{acc}$ is likely due to entrainment of particles from the residual layer, the decrease in the afternoon hours can be attributed to effective deposition in the forest canopy, as also discussed in Ahlm et al. (2009). The absorption coefficient at 637 nm, $\sigma_{ap\ 637}$, which is mostly




related to BC, follows a diel pattern (Fig. 5c-d) similar to the $N_{acc}$ trend for both seasons. Since BC is usually not emitted by near-by sources and it is generally transported in the accumulation mode, the

similarities with $N_{acc}$ diel patterns were expected. However, the wet season diel cycle of $\sigma_{ap\,637}$ exhibits a decreasing tendency that starts two hours earlier than the decrease in $N_{acc}$. This difference can be explained by the fact that $\sigma_{ap\,637}$ and $N_{acc}$ are mass and number-related measurements, respectively. Therefore, a size-dependent deposition would affect mass and number-related aerosol properties in a different way. This difference was more evident in the wet season when BC concentrations were not as

dominant as in the dry season. The diel pattern of BrC contribution during the dry season is significantly different from the $\sigma_{ap\,637}$ (BC) pattern. Brown carbon absorption at 370 nm, $\sigma_{ap\,BrC\,370}$, shows its highest values between 12:00 and 14:00 UTC (08:00 – 10:00 LT) in the dry season and starts decreasing at 14:00 UTC (10:00 LT), earlier than $\sigma_{ap\,637}$ and $N_{acc}$ (Fig. 5e). This observation implies that the BrC aerosol particles measured at the ATTO site are mixed down into the boundary layer in the

early morning and are then quickly photo-degraded during the day (Forrister et al., 2015; Wang et al., 2016b; Wong et al., 2017). This pattern is not observed during the wet season, when $\sigma_{ap\,BrC\,370}$ exhibits no significant diel variability.

Other remote site observations have found no significant diel variation of the absorption coefficient, due to efficient mixing of the planetary boundary layer (PBL) and low anthropogenic emissions (Chi et al.,

2013). However, the high convectivity at tropical latitudes makes possible the entrainment of high altitude air masses that bring regional and/or LRT emissions, as observed before at an Amazonian site during the dry season (Brito et al., 2014).

### 3.4 BC to CO enhancement ratio

Agricultural clearing fires, like savanna fires, are dominated by the flaming combustion phase, in

contrast to deforestation fires, where less than 50 % of the biomass is burned in the flaming phase (Dubovik et al., 2002). An important part of forest fires occurs in the form of smoldering combustion due to higher fuel moisture (Guyon et al., 2005). Under smoldering fire regimes, when the combustion is less efficient and thus, tends to emit more CO, observations tend to show lower $ER_{BC}$ and higher single scattering albedo, $\omega_0$, as well as higher organic carbon (OC) enhancement ratio, $ER_{OC}$. On the



other hand, flaming fires, which produce abundant BC aerosol particles, tend to exhibit lower $\omega_0$ and

higher $ER_{BC}$ (Akagi et al., 2011). The smoke that arrives at the ATTO site during the dry season is a

mixture of smoldering and flaming emissions with varying relative fractions. The air mass origin, (i.e.,

the back trajectories) largely defines if emissions are advected from regions with predominantly

smoldering or flaming fires (Pöhlker et al., 2017a).

The calculated $ER_{BC}$ values and $\omega_0$ allow us to distinguish between flaming and smoldering smoke and

locate the different sources. Figure 6 shows the $ER_{BC}$ and $\omega_0$ values measured at the ATTO site

classified by grouped BT clusters. It can be observed that mainly the ESE and E trajectory clusters have

$ER_{BC}$ higher than 8 ng m$^{-3}$ ppb$^{-1}$. From the two predominant BT cluster groups in the dry season (ESE

and E), the ESE trajectories seem to be the more influenced by flaming fires since the measurements are

more shifted to high $ER_{BC}$ and lower $\omega_0$. In fact, the ESE clusters are dominated by the $0.80 - 0.90$

$\omega_0$-range, which means they are highly loaded with light-absorbing aerosol. This evidence is supported

by the land cover information, which indicates that agricultural lands account for $6 - 20$ % of the ESE

clusters total land cover, $3 - 5$ % of the E clusters, and $< 1$ % of the ENE and NE clusters (Pöhlker et

al., 2017a). The eastern clusters (E) are more equally distributed in the $\omega_0$ range and tend to be lower in

terms of $ER_{BC}$ compared to the ESE clusters. Therefore, we expect E trajectories to be more influenced

by smoldering fires during the dry season compared to the ESE trajectories, even though, as already

mentioned, the arrival of African savanna fire smoke from easterly trajectories in August-September

provides BB aerosol particles that have lower $\omega_0$ and higher $ER_{BC}$.

During the wet season, when ENE and NE BT clusters are dominant, we observed a trend towards

lower $ER_{BC}$ and higher $\omega_0$ than expected, since the frequency of regional fires is much lower than in the

dry season. Actually, when including data from the beginning of 2016, under the influence of ENSO,

we observed a shift towards higher $ER_{BC}$ in the NE directions due to the occurrence of fires in the

Guyanas area. These atypical data were excluded from Fig. 6 to improve the contrast between the

different air mass trajectory clusters. The NE and ENE trajectories were very similar in terms of $\omega_0$ and

$ER_{BC}$. Occasional dust transport events from the Sahara, mixed with BB aerosol from the Sahel region,

brought aerosol particles with lower $\omega_0$ compared to the wet season average.




The lower $ER_{BC}$ observed in the wet season was likely due to aerosol scavenging during the transatlantic advection (Moran-Zuloaga et al., 2017), while CO is not affected by wet deposition (Liu et al., 2010).

One important aspect worth mentioning is the fact that $ER_{BC}$ decreased more steeply with increasing $\omega_0$

and their correlation was closer during the dry season (E and ESE BT clusters) in comparison to the wet season. This feature might be related to the age of the aerosol particles, because aging would make the BC become less hydrophobic (Pöhlker et al., 2017b) so that it can be more efficiently removed by wet scavenging.

**3.5 El Niño impact on aerosol optical properties**

The aerosol optical properties measured at ATTO changed during the El Niño period at the end of 2015 and the beginning of 2016 (Fig. 3). To have a broader view on how this phenomenon affected the Amazon rain forest aerosol, we added scattering and absorption data from ZF2 published in Rizzo et al. (2013) and extended with recent data to the current ATTO time series in Fig. 7a-b. Overlapping data in 2013 (Fig. 7a and 7b) are statistically equivalent with only a few days affected by probable near-site

sources. Positive Oceanic Niño Index (ONI) values (Fig. 7c) were found to be related to higher scattering and absorption coefficients in the dry season. However, the ENSO is not the only cause of precipitation anomalies in the Amazon Basin. The Atlantic Multi-Decadal Oscillation (AMO) has also been found to be causing droughts (Aragão et al., 2008). The non-ENSO average (ZF2 and ATTO) scattering coefficient at 637 nm during the dry seasons was $24 \pm 18$ Mm$^{-1}$. This average increased up to

$54 \pm 39$ Mm$^{-1}$ and $42 \pm 24$ Mm$^{-1}$ during the dry seasons 2009 and 2015, respectively. The wet season scattering coefficient average was also affected during El Niño, increasing from a non-ENSO average of $7 \pm 7$ Mm$^{-1}$ to $10 \pm 11$ Mm$^{-1}$ during the wet season 2016. A similar pattern was observed for $\sigma_{ap\ 637}$, which increased from a non-ENSO average in the dry seasons of $3.8 \pm 2.8$ Mm$^{-1}$ to $5.5 \pm 2.8$ Mm$^{-1}$ and $5.2 \pm 2.1$ Mm$^{-1}$ during the dry seasons in 2009 and 2015, respectively. It is remarkable that high

absorption coefficients were also measured during the dry season 2010 ($5.6 \pm 4.7$ Mm$^{-1}$), a year with mostly negative ONI. However, it has been shown that an increased sea surface temperature in the Atlantic Ocean (not ENSO related) in 2010 caused a special drought period in the Amazon rain forest (Lewis et al., 2011).



### 3.6 Absorption wavelength dependence and BrC contribution

Open biomass burning emits a mixture of BC and OA with high absorption wavelength dependence (Andreae and Gelencsér, 2006; Hoffer et al., 2006; Kirchstetter et al., 2004). However, our observations show that sometimes LAC measured at the ATTO site can fall in the BC-only regime, with $å_{abs} \approx 1$. To understand this pattern, we have analyzed the relationship between the WDA and other parameters, like the OA-to-sulfate ratio and $\omega_0$. In Fig. 8a, WDA is presented as a function of the OA-to-sulfate mass

ratio. According to the result of an orthogonal fit (not shown), there is a significant correlation between these variables ($R^2 = 0.61$), and the aerosol light absorption is in the BC-only regime (shaded area in Fig. 8a) when the OA-to-sulfate ratio is lower than ~6.5, which occurred 15 % of the time in the high-absorption periods ($\sigma_{ap\ 637}$ higher than the 75$^{th}$ percentile). On the other hand, higher OA-to-sulfate ratios correspond to likely BrC-rich aerosol masses, which were the majority of the cases. The $\omega_0$ at

637 nm of the BC-only regime (inter-quartile range, IQR: 0.82 – 0.86) was clearly lower than the one corresponding to the BrC-rich regime (IQR: 0.85 – 0.90).

In Fig. 8b, the BC-only regime data has been segregated by trajectory cluster. The air masses that are more likely to bring wavelength independent BC to the site are those with the faster wind speed: E3, E4, and ESE3. These emissions could be related to ship traffic in the Atlantic Ocean, BB in southern Africa,

or power plant emissions from the west African coast. Low OA-to-sulfate ratios with high $\omega_0$ occurred a few times and could be explained by high sulfate input from volcanic emissions in the Congo (Fioletov et al., 2016; Saturno et al., 2017a), rather than fossil fuel emissions, which are typically rich in BC.

In an effort to identify the BrC-rich trajectories, the WDA was studied for the different BT clusters that are mostly active during the dry season. Boxplots corresponding to each trajectory cluster, together with

the average fire counts in the geographical cluster area, are presented in Fig. 9. From the group of ESE trajectory clusters (ESE1, ESE2, and ESE3), the ESE1 trajectories exhibit the highest WDA values, with a decreasing tendency towards faster trajectories, ESE3 being the one with lowest WDA values. Even though ESE3 is the trajectory cluster with more fire counts, the fact that those fires occur farther from the ATTO site compared to the ones in the slowest trajectory, ESE1, could be related to a decrease

in absorption wavelength dependence during transport. A similar pattern is observed for the easterly



trajectory clusters (E1, E2, E3, and E4), where the slowest air mass trajectories comprised by the E1 cluster exhibit the highest WDA median compared to the rest of the E clusters. In the case of E4, the WDA 25$^{th}$ percentile is lower than the rest of the E trajectories, but it shows an increased median that can not be explained by the occurrence of fire events, which is lower than the observations for the other

clusters (E2, E3, and E4). The E4 weighted fire counts is anyhow in the same order magnitude as E2 and E3 and the wavelength dependence differences could be related to different fuel types or combustion phase. Actually, the long E clusters (E3 and E4) cover more southern areas than the shorter ones (E1 and E2) and have some overlap with ESE3. By comparing grouped E and ESE clusters, it can be observed that WDA in the E clusters has higher variability compared to the ESE ones. This pattern

could be associated with a wider range of sources in the E trajectories compared to ESE. The E trajectories travel over the Amazon River where ship traffic is quite significant. In fact, as can be observed in Fig. 9, for the E3 and E4 trajectories, there is a significant amount (> 25$^{th}$ percentile) of measurements that fall in the BC-only regime. Something similar is only observed for the ESE3 trajectories among the ESE group. Most of the agricultural land is located along the southern margins of

the Amazon rain forest (Pöhlker et al., 2017a). This area is within the ESE clusters footprint. The narrower range of WDA values measured for the ESE trajectories compared to the E ones, indicates that sources in the ESE footprint are more homogeneous compared to the sources located in the E footprint. These WDA tendencies could be useful to understanding the BrC emissions and atmospheric transformations in the context of the Amazon rain forest and its surroundings.

Using the calculated BC-only WDA thresholds, we were able to estimate the BrC contribution to total absorption during the measurement period (2012 – 2017) (Fig. 10). We found that BrC contributes 24 % (IQR: 17 – 29 %) of total light absorption at 370 nm wavelength. A slight seasonal variability was observed for the BrC relative contribution, with the medians and IQR during the wet and dry season being 27 % (19 – 34) and 22 % (16 – 27), respectively. However, most of the wet season data had to be

excluded, because they were from air masses rich in mineral dust, which introduces large uncertainties in the WDA method. During El Niño, at the end of 2015, open fire events were more frequent (with weighted fire counts of 1756 km$^{-2}$ compared to the 2008 – 2016 average of 1076 km$^{-2}$), and the CO 95$^{th}$ percentile was exceeded several times. In this period, the BrC contribution had a median of 37 % (IQR:



27 – 47) and showed a significant correlation with CO ($R^2 = 0.47$). This significant increase of the BrC
contribution could be related to the relatively short distance between the fire spots and the ATTO site. It
can be observed in Fig. 10 that the El Niño influence continued during the dry season 2016 but not as
strongly as in 2015. Previous observations have shown that the atmospheric lifetime of BB-emitted BrC
is ~1 day due to photolysis and oxidation, which destroy the chromophores (Forrister et al., 2015; Wang
et al., 2016b; Wong et al., 2017). Therefore, BrC emitted from fires in the southern borders of the
Amazon rain forest, which require ~3 days to be transported to the ATTO site, is likely to be
significantly photodegraded and to contribute only weakly to total aerosol light absorption after
atmospheric processing.

The BC to OA mass ratio during the sampling time had a median of 0.06 (IQR: 0.04 – 0.10). The ratio
BC to OA has been used before to parameterize $å_{abs}$ and $\omega_0$ (Pokhrel et al., 2016; Saleh et al., 2014), but
little is known about this relationship for tropical forest emissions. A broader range of BC to OA mass
ratio between 2014 and 2016 was observed during the dust episodes in the wet season, including those
periods when regional fires were active (IQR: 0.08 – 0.24). Other periods, like the dry season, with
higher BC mass concentrations, exhibited a narrower and lower BC to OA mass ratio range
(IQR: 0.03 – 0.08). A scatter plot of the absorption wavelength dependence, $å_{abs}$, as a function of the BC
to OA mass ratio during the North African LRT events in the wet season can be found in Fig. 11. We
have found a trend where $å_{abs}$ increases with decreasing BC to OA mass ratio following an exponential
function. These results are comparable to those presented by Pokhrel et al. (2016) and Saleh et al.
(2014), with slightly lower $å_{abs}$ values in our study, however. This pattern could be related to a dominant
presence of primary organic aerosol (POA) that has characteristically lower absorption wavelength
dependence compared to SOA (Saleh et al., 2013). However, more experimental studies are required to
investigate the optical properties of aerosol produced by burning different tropical forest fuels.

**Summary and conclusions**

The optical properties of aerosol particles sampled at the ATTO site have been presented for a
measurement period of 5 years (2012 – 2017). Seasonal trends affected light scattering and absorption



by aerosol particles, showing a significant increase during the dry season due to a higher frequency of regional open fires. The wet season was dominated by background biogenic aerosol, occasionally disrupted by LRT dust and BB aerosol transported from Africa to the ATTO site, which lead to decreased $\mathring{a}_{sca}$ and $\omega_0$ (637 nm). The average $\omega_0$ during the wet season was 0.93 ± 0.04, higher than the dry season average of 0.87 ± 0.03. The absorption wavelength dependence, $\mathring{a}_{abs}$, was relatively low with

an average of 0.93 ± 0.16, and only slight variations between seasons. The highest $\mathring{a}_{abs}$ were measured in the presence of BB events but no effect on $\mathring{a}_{abs}$ was observed due to the presence of dust, most likely due to a size effect, given that absorption coefficients were measured only for sub-micron aerosol particles after May 2014. Black carbon MAC at 637 nm calculated from MAAP and SP2 measurements was comparable to other studies, although higher than "typical" values commonly used in the literature to

convert $\sigma_{ap}$ into BC mass concentrations. The calculated wet season MAC average was 11.4 ± 1.2 m² g⁻¹, while during the dry season the MAC average was increased slightly to an average of 12.3 ± 1.3 m² g⁻¹ at 637 nm. These values are consistent with a strong "lensing effect" by organic coatings attached to BC aerosol particles. High OA amounts in the Amazonian atmosphere resulted in low BC to OA mass ratios, which were in the range of 0.04 to 0.10 (IQR). A significant correlation between BC to OA mass

ratio and $\mathring{a}_{abs}$ was observed during the wet season under the influence of regional and remote BB emissions. The ΔBC/ΔCO enhancement ratios (ER$_{BC}$) were mostly lower than 8 ng m⁻³ ppb⁻¹, mainly due to the aging of BB aerosol particles during transport to the site. A higher and wider range of ER$_{BC}$ values was observed during the dry season due to the influence of different biomass combustion phases that varied from smoldering to flaming fires.

Theoretical wavelength-dependent BC $\mathring{a}_{abs}$ were calculated and used to estimate the BrC contribution to near-UV (370 nm) light absorption. This approach resulted in medians of 27 and 22 % BrC contributions in the wet and dry season, respectively. Higher BrC contributions were measured during the El Niño season at the end of 2015. During this period, the BrC absorption at 370 nm increased to a median of 37 %. We observed that winds coming from ESE directions in the dry season were more

likely to bring aerosols with a high absorption wavelength dependence, implying a higher BrC content.

In the case of prolonged drought periods in the Amazon Basin, significant increases of BrC absorption contribution could be expected due to increased fire occurrence. Long-term monitoring of light



absorbing aerosol particles is required to reduce uncertainty in global climate models. The data presented here provide a contribution in this direction and can help to understand how different climatic

phenomena, like El Niño, can affect the Amazon atmospheric aerosol cycling. Further investigations on the BC mixing state and morphology will be required to improve modeled calculations and BrC retrievals.

# Acknowledgements

This work has been supported by the Max Planck Society (MPG) and the Max Planck Graduate School (MPGS). For the operation of the
ATTO site, we acknowledge the support by the German Federal Ministry of Education and Research (BMBF contract 01LB1001A) and the Brazilian Ministério da Ciência, Tecnologia e Inovação (MCTI/FINEP contract 01.11.01248.00) as well as the Amazon State University (UEA), FAPEAM, LBA/INPA and SDS/CEUC/RDS-Uatumã. P. A. acknowledges support from FAPESP – Fundação de Amparo à Pesquisa do Estado de São Paulo. J. S. is grateful for the PhD scholarship from the Fundación Gran Mariscal de Ayacucho (Fundayacucho). This paper contains results of research conducted under the Technical/Scientific Cooperation Agreement between the
National Institute for Amazonian Research, the State University of Amazonas, and the Max-Planck-Gesellschaft e.V.; the opinions expressed are the entire responsibility of the authors and not of the participating institutions. We highly acknowledge the support by the Instituto Nacional de Pesquisas da Amazônia (INPA). We would like to especially thank all the people involved in the technical, logistical, and scientific support of the ATTO project, in particular Reiner Ditz, Jürgen Kesselmeier, Alberto Quesada, Niro Higuchi, Susan Trumbore, Matthias Sörgel, Thomas Disper, Andrew Crozier, Uwe Schulz, Steffen Schmidt, Antonio Ocimar Manzi, Alcides Camargo
Ribeiro, Hermes Braga Xavier, Elton Mendes da Silva, Nagib Alberto de Castro Souza, Adi Vasconcelos Brandão, Amaury Rodrigues Pereira, Antonio Huxley Melo Nascimento, Feliciano de Souza Coehlo, Thiago de Lima Xavier, Josué Ferreira de Souza, Roberta Pereira de Souza, Bruno Takeshi, and Wallace Rabelo Costa.





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





**Table 1.** List of frequently used symbols and acronyms

| Description | Acronym | Symbol | Units |
|---|---|---|---|
| Black carbon | BC | | |
| Brown carbon | BrC | | |
| Equivalent black carbon | $BC_e$ | | |
| Refractory black carbon | rBC | | |
| Organic carbon | OC | | |
| Organic aerosol | OA | | |
| Light-absorbing carbonaceous matter | LAC | | |
| ΔBC/ΔCO enhancement ratio | $ER_{BC}$ | | |
| Attenuation coefficient | ATN | $\sigma_{ATN}$ | $m^{-1}$ |
| Absorption coefficient | | $\sigma_{ap}$ | $m^{-1}$ |
| Scattering coefficient | | $\sigma_{sp}$ | $m^{-1}$ |
| Absorption Ångström exponent | AAE | $\mathring{a}_{abs}$ | |
| Scattering Ångström exponent | SAE | $\mathring{a}_{sca}$ | |
| Wavelength dependence of $\mathring{a}_{abs}$ | WDA | | |
| Mass attenuation cross-section | | $\alpha_{atn}$ | $m^2\,g^{-1}$ |
| (BC) Mass absorption cross-section | MAC | $\alpha_{abs}$ | $m^2\,g^{-1}$ |
| Backscattering coefficient | | $\sigma_{bsp}$ | $m^{-1}$ |
| Single scattering albedo | SSA | $\omega_0$ | |
| Aerosol optical depth | AOD | | |
| Condensation nuclei number concentration (> 10 nm) | | $N_{CN}$ | $cm^{-3}$ |
| Accumulation mode particle number concentration (100 to 430 nm) | | $N_{acc}$ | $cm^{-3}$ |
| Precipitation at ATTO region of interest (ROI), Fig. 1a | | $P_{ATTO}$ | mm |
| Equivalent potential temperature | | $\theta_e$ | K |
| Amazon Tall Tower Observatory | ATTO | | |
| Backward trajectory | BT | | |
| Long-range transport | LRT | | |
| El Niño Southern Oscillation | ENSO | | |
| Oceanic Niño Index | ONI | | |
| Biomass burning | BB | | |
| Fossil fuel | FF | | |
| Coordinated universal time | UTC | | |
| Local time | LT | | |
| Inter-quartile range | IQR | | |
| Domain of interest, Fig. 2a | DOI | | |





**Table 2.** Descriptive statistics (mean ± standard deviation and interquartile range, IQR) of daily-averaged aerosol optical properties over the Amazon rain forest during the different seasons and the entire measurement period.

| | Wavelength | Wet season (Feb – Mar – Apr – May) | | Dry season (Aug – Sep – Oct – Nov) | | Entire period (2012 – 2017) | |
|---|---|---|---|---|---|---|---|
| | | Mean ± SD | IQR | Mean ± SD | IQR | Mean ± SD | IQR |
| Scattering coefficient $\sigma_{sp}$ [Mm$^{-1}$] | 450 nm | 9 ± 11 | (5.1 – 11) | 47 ± 35 | (24 – 64) | 31 ± 35 | (10 – 39) |
| | 550 nm | 7.5 ± 9.3 | (3.8 – 8.7) | 33 ± 25 | (17 – 46) | 22 ± 25 | (7 – 28) |
| | 637 nm | 6.4 ± 8.9 | (3.0 – 7.4) | 26 ± 19 | (13 – 35) | 17 ± 19 | (6 – 23) |
| Absorption coefficient $\sigma_{ap}$ [Mm$^{-1}$] | 637 nm | 0.68 ± 0.91 | (0.17 – 0.72) | 4.0 ± 2.2 | (2.4 – 5.1) | 2.1 ± 2.2 | (0.43 – 3.2) |
| Single scattering albedo $\omega_0$ | 637 nm | 0.93 ± 0.04 | (0.91 – 0.96) | 0.87 ± 0.03 | (0.84 – 0.89) | 0.89 ± 0.04 | (0.86 – 0.93) |
| Scattering Ångström exp. * $å_{sca}$ | | 1.29 ± 0.50 | (0.98 – 1.65) | 1.71 ± 0.24 | (1.53 – 1.89) | 1.54 ± 0.42 | (1.32 – 1.84) |
| Absorption Ångström exp. * $å_{abs}$ | | 0.91 ± 0.19 | (0.80 – 0.98) | 0.94 ± 0.16 | (0.84 – 1.03) | 0.93 ± 0.16 | (0.83 – 1.01) |
| Mass absorption cross-section $\alpha_{abs}$ [m$^2$ g$^{-1}$] ** | 637 nm | 11.4 ± 1.2 | (10.8 – 12.0) | 12.3 ± 1.3 [a] | (11.4 – 13.3) [a] | 11.9 ± 1.4 | (11.0 – 13.0) |

* Calculated by applying a log-log linear fit to measurements at all available wavelengths.
** Calculated by fitting 8-channel SP2 and MAAP data.
[a] Including data from July 2015/16 (wet-to-dry transition season).





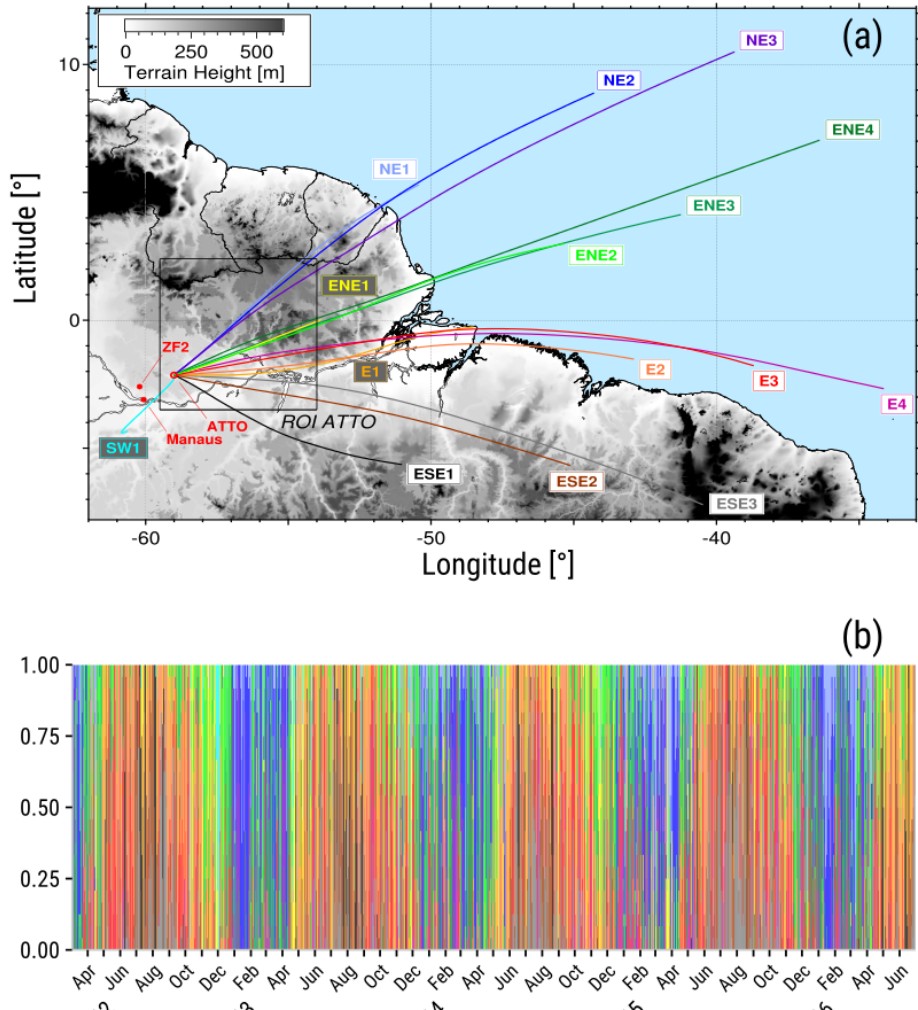

**Figure 1.** (a) Map of the northeastern Amazon Basin including averaged backward trajectory clusters and the region of interest (ROI) (59° W to 54° W; 3.5° S to 2.4° N), as a black rectangle, used to retrieve precipitation in the ATTO area. (b) Time series of the frequency of occurrence (FoO) of each BT cluster during the sampling period. Adapted from Pöhlker et al. (2017).





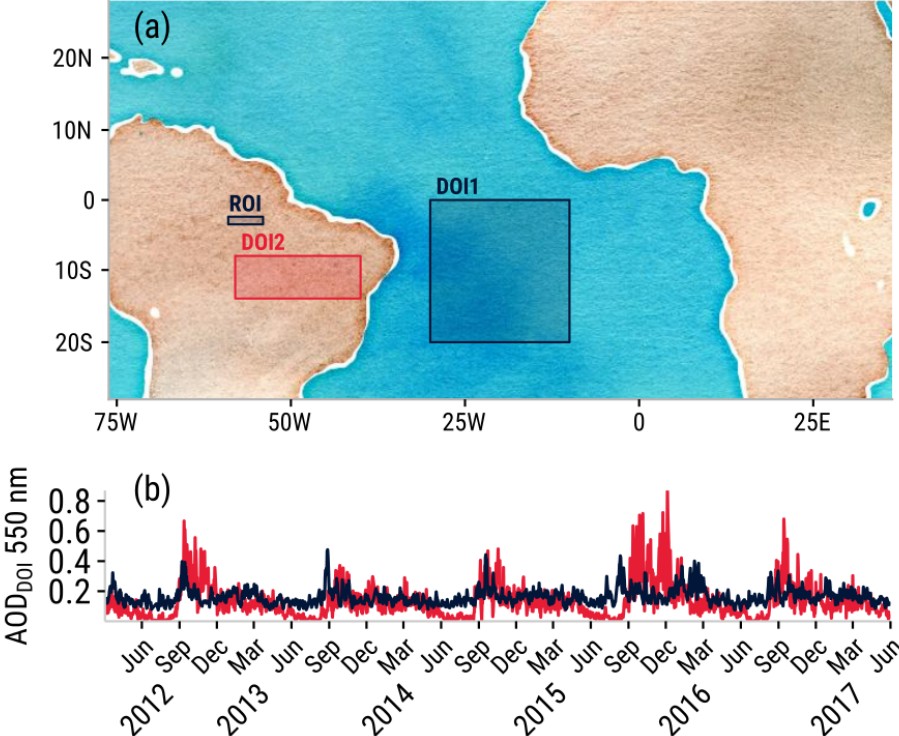

**Figure 2.** Aerosol optical depth (550 nm) observations in two different domains of interest (DOI1 and DOI2), as shown in (a). Time series of area-averaged AOD are shown in (b) for DOI1 (dark blue) and DOI2 (red). The ATTO region of interest (ROI) is shown as a black rectangle in (a).





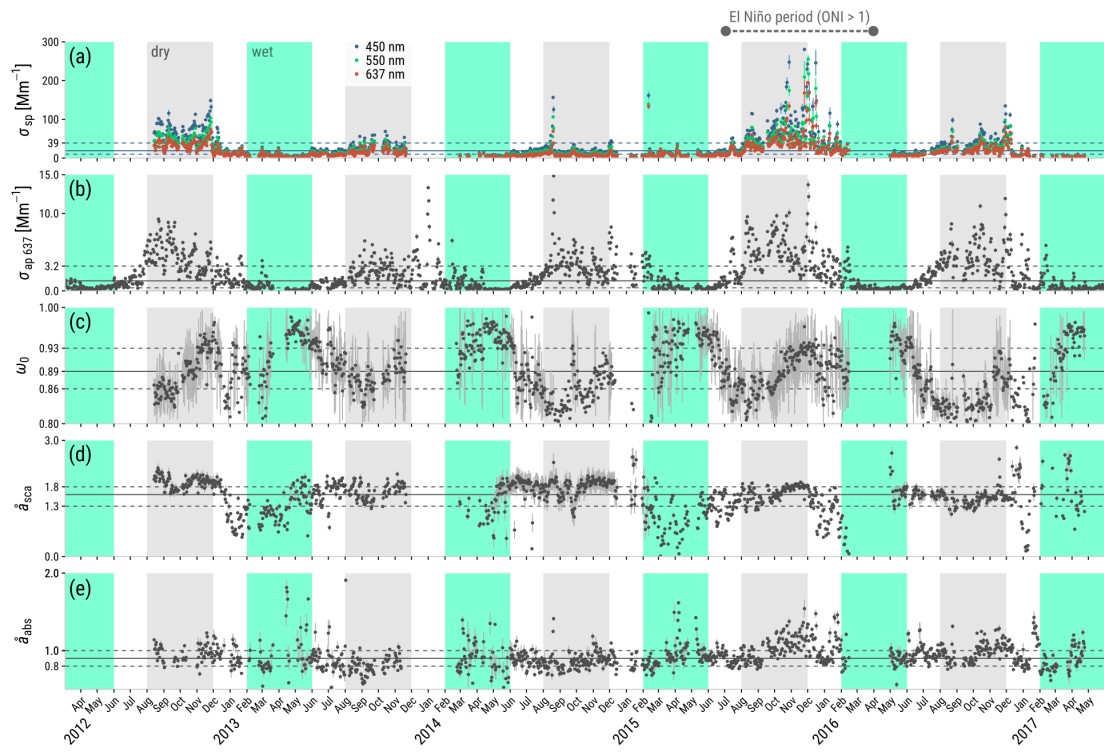

**Figure 3.** Overview of aerosol optical properties during the measurement period. (a) Scattering coefficient, (b) absorption coefficient at 637 nm, (c) single scattering albedo at 637 nm, (d) scattering Ångström exponent, and (e) absorption Ångström exponent. All data were averaged on 24-h intervals and standard errors are presented as vertical gray bars. Green and gray shaded areas correspond to the wet and dry seasons, respectively. First and third quartiles are represented as horizontal dashed lines, and medians as horizontal solid lines.



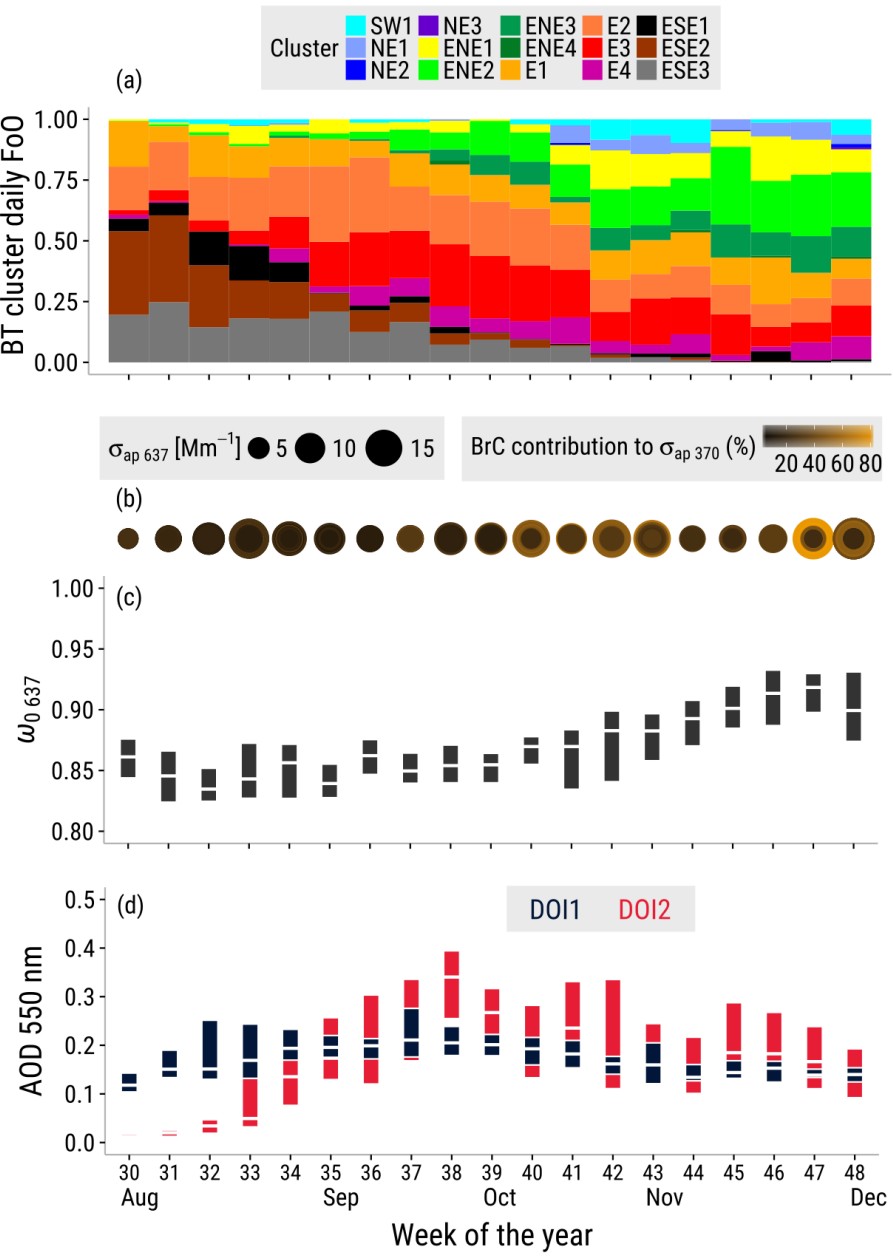

**Figure 4.** Multi-year (2012 – 2017) weekly averages over the dry season corresponding to (a) frequency of occurrence of BT clusters, (b) absorption coefficients at 637 nm ($\sigma_{ap\,637}$) shown as circles with increasing diameters, the color scale corresponds to the relative BrC contribution to $\sigma_{ap\,370}$, (c) single scattering albedo at 637 nm ($\omega_{0\,637}$), and (d) aerosol optical depth at 550 nm (AOD) for the different domains of interest, DOI1 and DOI2, which cover regions of the South Atlantic Ocean and the southern Amazon, respectively. Boxplots in (c) and (d) represent the median (white segment) and the 25th and 75th percentiles (lower and upper box edges, respectively).





**Figure 5.** Diel variation of (a, b) median of the accumulation mode particle number concentration, $N_{acc}$, (c, d) median of the absorption coefficient at 637 nm, (e, f) median of the BrC absorption coefficient at 370 nm, (g, h) precipitation rate, and (i, j) median of the equivalent potential temperature. Gray and white backgrounds correspond to the night and day times, respectively. Error bars correspond to the standard error. Please note the different $y$-axis scales in (a-f).





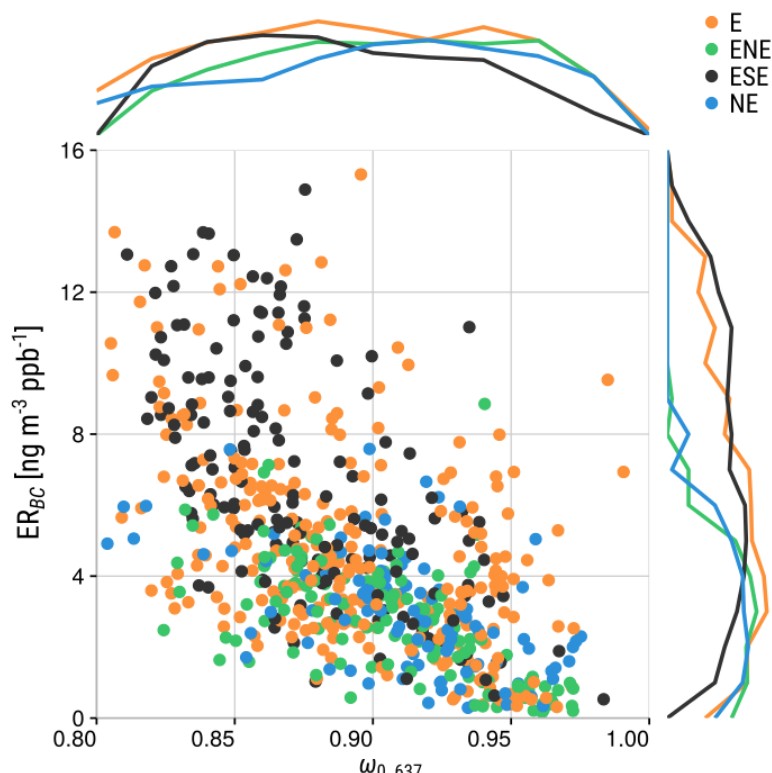

**Figure 6.** Scatter plot of the black carbon to CO enhancement ratio ($ER_{BC}$) vs. single scattering albedo at 637 nm ($\omega_{0\ 637}$) with marginal probability density plots (normalized counts in log-scale) for data corresponding to grouped back-trajectory clusters. Each point represent a 24-h average.





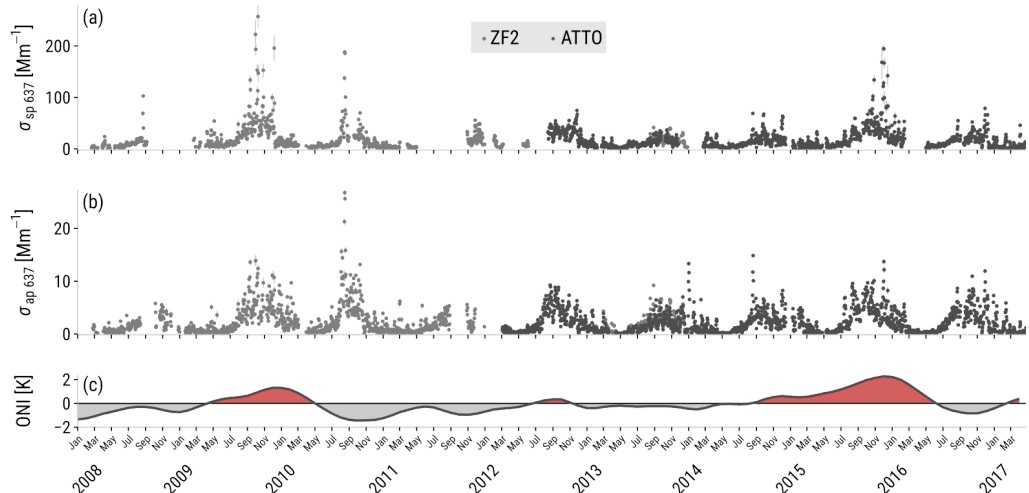

**Figure 7.** Scattering (a) and absorption (b) coefficient (637 nm) time series measured at the ZF2 and the ATTO sites from 2008 to 2016 (24-h averaged data). Increased scattering and absorption coefficients were observed under the influence of El Niño. (c) High ONI indicates active ENSO periods, shown as red shaded areas.





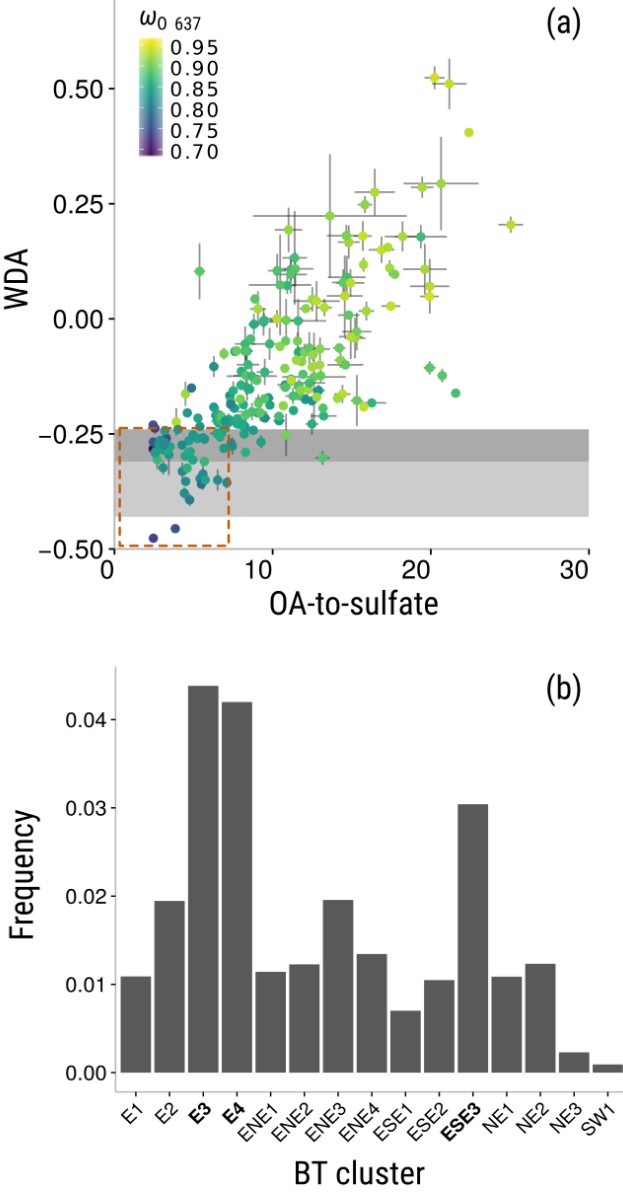

**Figure 8.** (a) Absorption wavelength dependence (WDA) as function of the OA-to-sulfate mass ratio during high-absorption periods in the dry season. The color scale indicates the $\omega_0$ at 637nm. Gray shaded areas correspond to theoretical WDA for internally mixed BC (light gray), and externally mixed BC (dark gray). The data inside the dashed rectangle in (a) is used in (b) to identify the BT clusters that are more likely to bring BC to the ATTO site.





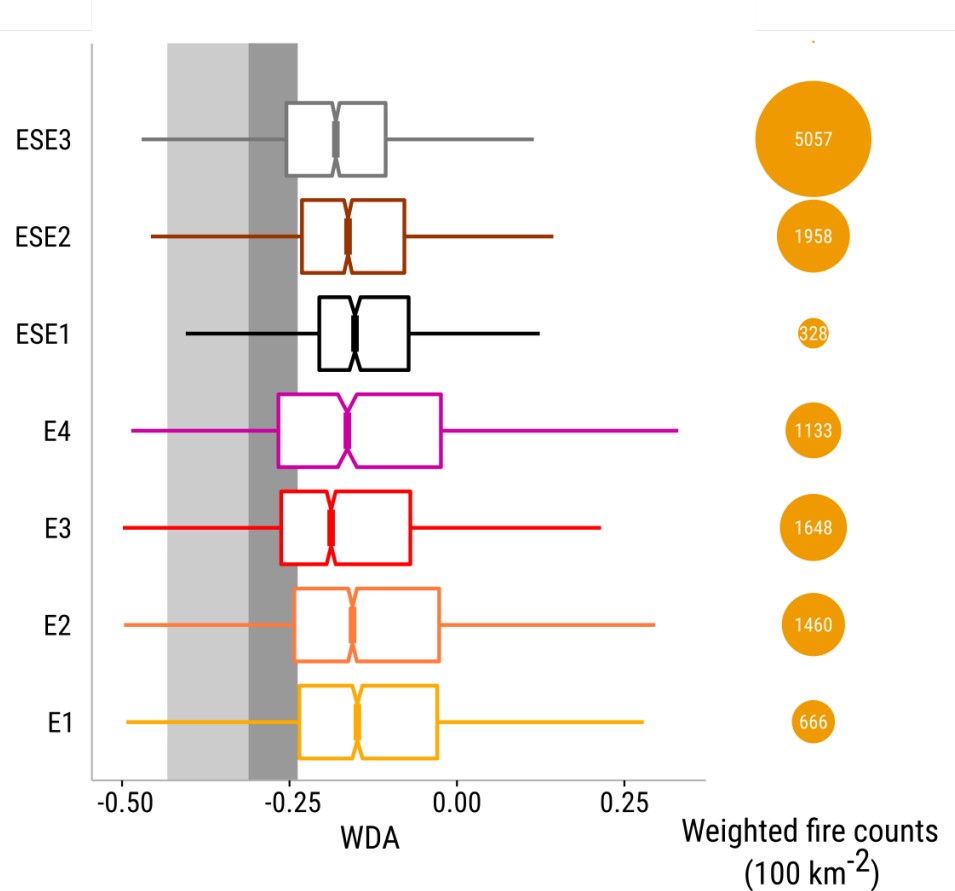

**Figure 9.** Wavelength dependence of $\mathring{a}_{abs}$ (WDA) for different trajectories in the dry season presented as box and whisker plots (left). The light and dark gray shaded areas correspond to the pure BC and internally mixed BC regimes, respectively. Notches correspond to 1.58 IQR $n^{-\frac{1}{2}}$. If notch ranges do not overlap, the medians are statistically different (95% confidence). The trajectory weighted fire counts for each BT cluster are shown as circles on the right side. The data presented here correspond to 1-h averages.





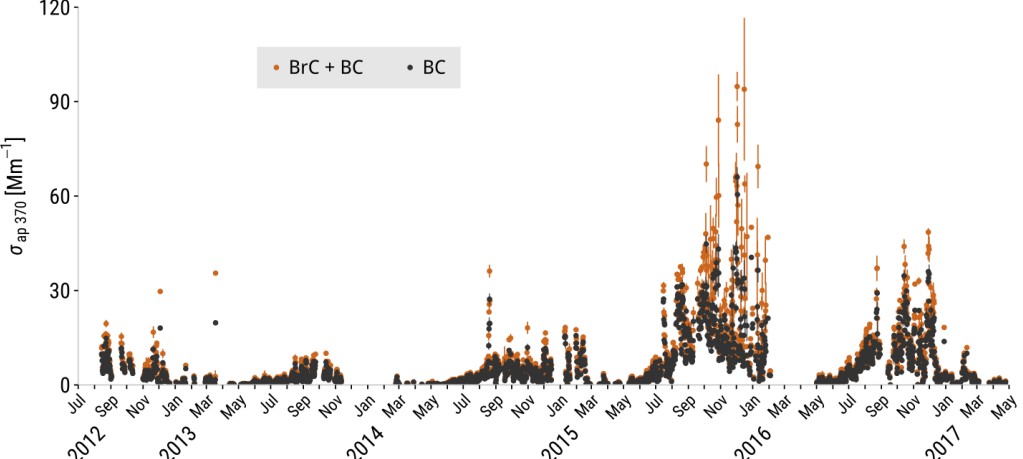

**Figure 10.** Total absorption at 370 nm (12-h average data) segregated by BC only (gray points) and BrC + BC (brown points). Error bars are equivalent to ± 1 standard error. Long-range transport dust events have been excluded from the analysis.





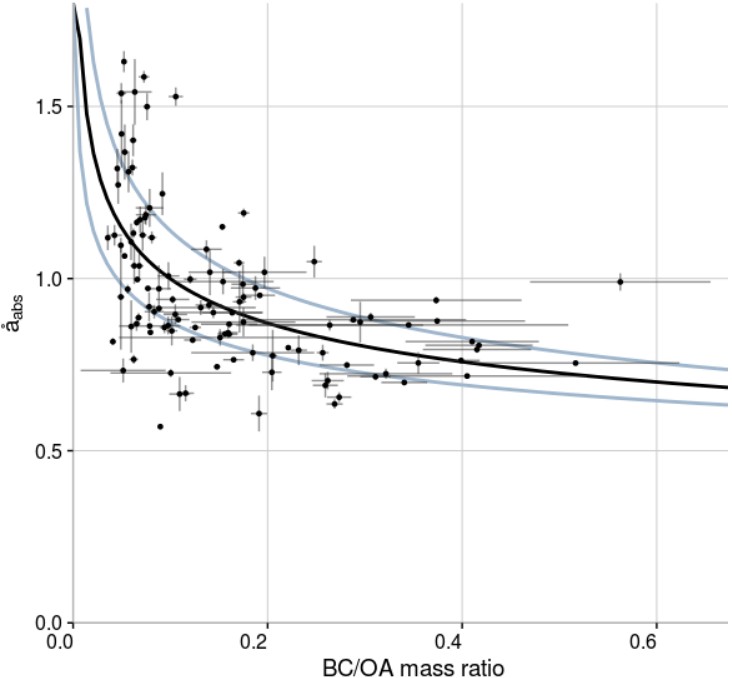

**Figure 11.** Absorption Ångström exponent ($\mathring{a}_{abs}$) as a function of the BC/OA mass ratio for selected dust events in the wet season. The black line corresponds to a non-linear least squares fit applied to the data ($y = x^{-0.199} \times 0.632$). The light blue lines correspond to the standard error of the fit.