# Peer review of "Black and brown carbon over central Amazonia: Long-term aerosol measurements at the ATTO site"

_Atmospheric Chemistry and Physics, 2017_

## Referee Comment (RC1) · Anonymous Referee #1 · 31 Jan 2018

At first, I want to apologise for the delay of my review.

The paper investigates the occurrence and optical properties of black and brown carbon during a five-year period based on ground-based observations at the ATTO site in the Amazon forest. In particular, the impact of different airmass dynamics and El Nino conditions on the optical properties and relative contribution of black and brown species is investigated. I find the paper well structured and clearly written, and the data presented of great value. In my opinion the paper deserve publication only after minor revisions. Main comments are detailed in the following.

Comments

[Figure]

The introduction is quite long and introduces to many concepts. I do not know if it would be better to split it in sub-paragraphs giving a theoretical background on the topic. Anyhow, it is a good state-of-the art of black and brown carbon studies.

Page 14, line 362 : what do you mean with characteristic size distribution ? the average size ? Please be more precise.

Section 3.1, page 19, line 482 : you state that dry and wet periods are related to different aerosols influences (biomass burning and dust/sea salt respectively). However different signatures are not present in the temporal absorption angstrom exponent (Fig. 3, panel e). Can you comment on this point ?

Line 492-526 : probably this part can be moved in a single paragraph focusing on the MAC

Line 529-530 : many time the authors state, but do not prove, that the dry season is affected by BB particles. Is this assumption made based on previous studies at the site ? The same for the dust influence during the wet season. I think this point should be better addressed in the paper before analyzing in more detail the optical properties of the different aerosol types in different periods. This was the only part that I found not clear at all in the paper.

Line 565 : you state that is the contribution of sulfate that increases scattering. Why not the mixing with other compounds or species ?

Lines 568-570 : how do you select and eliminate from the dataset the BB and mineral dust events, and what "extreme event" means (AOD higher than a threshold ?). Please be more precise.

---

## Referee Comment (RC2) · Anonymous Referee #2 · 6 Mar 2018

Saturno et al., acp-2017-1097

REVIEW

GENERAL
The paper presents analyses of aerosol optical properties measured at the ATTO site in Amazonia during several years.  The authors have measured scattering and absorption coefficients, refractory BC (rBC), aerosol chemical composition as well as several supporting parameters. The wavelength dependence of absorption was used for estimating the contributions of black and brown carbon to light absorption, contributions of geographical source areas were estimated using transport analyses, mass absorption coefficients (MAC) by comparing independent absorption and rBC measurements. All this is important and valuable. The paper is mainly well written so I can recommed its publication after some clarifications and revision.

The main point that bothers me is the way the contribution of brown carbon is calculated. It is the core of the paper so it should be presented clearly. I'll show the problem in the detailed comments.

Another point that I miss is the size distributions of rBC. They were measured with the SP2 and used for MAC calculations but not shown anywhere. Why? The geometric mean diameters and geometric standard deviations of BC are useful and valuble as such for modeling purposes. Did they vary seasonally and with source areas? The size distríbution can also give hints of whether part of the BC remained undetected which would definitely affect the calculated MAC values, another important point in this paper. There should be some uncertainty analysis of the MAC.

How about coatings? They can be obtained from the SP2 but not presented, why? It would be valuable for the analysis of absorption enhancement.

DETAILED COMMENTS

L112 "(Womack et al., ref needed)."   Write the ref.

L221-227. I assume you corrected also the Aurora 3000 data for truncation, did you?

L234 "... instrument is able to provide absorption coefficients with a time resolution of 5 min."
The time resolution of the MAAP can be set not only to 5 min. Reword the sentence.

L 303-304 "... . The 8-channel SP2 rBC mass measurement was underestimated by a factor of 5 % ..."
How was this 5% obtained?
Another thing is, how could you estimate missing BC if it were outside the size range detected by the SP2? In biomass burning smoke BC could be attached to larger particles as well.

L356-372, calculation of BrC. There is a problem here. I rewrite the equations and show it.

$$WDA = \mathring{a}_{abs370-950} - \mathring{a}_{abs660-950} \tag{3}$$

$$BC\mathring{a}_{abs370-950} = \mathring{a}_{abs660-950} + WDA \tag{4}$$

$$BC\sigma_{ap370} = \sigma_{ap950} \times \left(\frac{370}{950}\right)^{-BC\mathring{a}_{abs370-950}} \tag{5}$$

$$BrC\sigma_{ap370} = \sigma_{ap370} - BC\sigma_{ap370} \tag{6}$$

Insert (3) to (4)

$$\Rightarrow BC\mathring{a}_{abs370-950} = \mathring{a}_{abs660-950} + WDA = \mathring{a}_{abs660-950} + \mathring{a}_{abs370-950} - \mathring{a}_{abs660-950} = \mathring{a}_{abs370-950}$$

Insert the result to (5)

$$\Rightarrow BC\sigma_{ap370} = \sigma_{ap950} \times \left(\frac{370}{950}\right)^{-BC\mathring{a}_{abs370-950}} = \sigma_{ap950} \times \left(\frac{370}{950}\right)^{-\mathring{a}_{abs370-950}} = \sigma_{ap370}$$

where the last step comes from applying (1) on line 342.
Insert finally the result to (6) and you get that $BrC\sigma_{ap370} = 0$.

This cannot be the idea, BrC being always zero. Rewrite the equations.

Further on the same issue. On L361-362 it is written " Calculated BC WDA thresholds, presented in Fig. S5, were compared to the ambient data in order to retrieve the BrC contribution to light absorption. "
What do you mean by thresholds? Do you mean that if WDA is larger than a threshold then this is due to BrC? This is not clear at all. Looking at Fig S5 does not explain me, what these thresholds might be. And what is the reasoning for claiming that exceeding a threshold for WDA is due to BrC?
This is much too descriptive way to explain how you calculated BrC. Give more details so that other people can us the same method and evaluate it. Recently many people have started using the so-called "Aethalometer model" (Sandradewi et al., 2008) and it would be good if the model presented in this work could be compared with it.

L373-377, still about the same issue. How did you get the uncertainties? Give formulas. Writing that
" The relative overestimation of the BrC contribution obtained by using different BC core sizes and different refractive indices in the Mie model calculations can be found in Table S2." is simply too qualitative an not understandable. People have to be able to reproduce the result.

L472-473 " Rizzo et al. (2013), however, pointed out that this relationship is only evident for surface and volume mean diameters and was not clearly valid between $\mathring{a}_{sca}$ and count mean diameters. "
Also Virkkula et al.: ACP, 11, 4445–4468, 2011 found the same.

L568 -> Where do you get the equivalent potential temperature from? Model, I assume, present in methods then.

L672 Do you have a clear definition for "BC-only regime"?

---

## Author Comment (AC1) · 2 Jun 2018

**Response to RC1**

We appreciate the reviewer's comments and suggestions that helped to improve the manuscript. Our responses are presented below, including the original comments from the reviewer, which are presented with gray background.

The paper investigates the occurrence and optical properties of black and brown carbon during a five-year period based on ground-based observations at the ATTO site in the Amazon forest. In particular, the impact of different airmass dynamics and El Nino conditions on the optical properties and relative contribution of black and brown species is investigated. I find the paper well structured and clearly written, and the data presented of great value. In my opinion the paper deserve publication only after minor revisions. Main comments are detailed in the following.

The introduction is quite long and introduces to many concepts. I do not know if it would be better to split it in sub-paragraphs giving a theoretical background on the topic. Anyhow, it is a good state-of-the art of black and brown carbon studies.

**AUTHORS**

We would like to thank the reviewer for the comments related to the introduction and the article in general. We have improved the introduction in order to make it more concise. Please see the revised version of the manuscript.

REVIEWER

Page 14, line 362 : what do you mean with characteristic size distribution ? the average size ? Please be more precise.

**AUTHORS**

We have included number and size distribution plots in the supplementary material to clarify the statements written in the manuscript.

REVIEWER

Section 3.1, page 19, line 482 : you state that dry and wet periods are related to different aerosols influences (biomass burning and dust/sea salt respectively). However different signatures are not present in the temporal absorption angstrom exponent (Fig. 3, panel e). Can you comment on this point ?

**AUTHORS**

The average of hourly mean $\mathring{a}_{abs}$ values measured at the ATTO site during this study was slightly higher during the dry season compared to the wet season, as stated in the original manuscript: "0.94 ± 0.16 compared to a wet season average of 0.91 ± 0.19." We hypothesized a more pronounced seasonality due to the larger occurrence of fires in the dry season bringing BrC-rich aerosol. However, we have found that these conditions occurred episodically rather than seasonally, with increased $\mathring{a}_{abs}$ only when ATTO was under strong influence of likely close-by biomass burning.

The following comment was added to section 3.1 of the revised manuscript:

"It was found that the $\mathring{a}_{abs}$ only increased significantly during episodes of hours or days, typically caused by nearby burning during the dry season".

Regarding the mineral dust effect on $\mathring{a}_{abs}$, no influence was found during the dust periods, as stated in the manuscript: "no effect on $\mathring{a}_{abs}$ was observed due to the presence of dust, most likely due to a size effect, given that absorption coefficients were measured only for sub-micron aerosol particles after May 2014".

**REVIEWER**

Line 492-526 : probably this part can be moved in a single paragraph focusing on the MAC

**AUTHORS**

Point taken; in the new version of the manuscript the MAC discussion is included in a new section.

REVIEWER

Line 529-530 : many time the authors state, but do not prove, that the dry season is affected by BB particles. Is this assumption made based on previous studies at the site? The same for the dust influence during the wet season. I think this point should be better addressed in the paper before analyzing in more detail the optical properties of the different aerosol types in different periods. This was the only part that I found not clear at all in the paper.

**AUTHORS**

Point taken. The influences of biomass burning and mineral dust over the Amazon rain forest are indeed well documented in the literature, therefore we have included the following references in the introduction of the revised manuscript:

Biomass burning influence during the dry season:

(Andreae et al., 1988; Artaxo et al., 2002; Fuzzi et al., 2007; Guyon et al., 2003; Roberts et al., 2003)

Saharan dust influence during the wet season:

(Formenti et al., 2001; Guyon et al., 2004; Moran-Zuloaga et al., 2017; Prospero et al., 1981; Talbot et al., 1990; Wang et al., 2016), already mentioned in the original manuscript.

REVIEWER

Line 565 : you state that is the contribution of sulfate that increases scattering. Why not the mixing with other compounds or species ?

**AUTHORS**

We agree with the reviewer and have modified the manuscript accordingly.

Original version:

> "In terms of the single scattering albedo ($\omega_0$, Fig. 4c), its increase towards the end of the dry season confirms that the aerosol particles during this time are scattering more radiation, not only due to higher BrC presence but also due to an increased sulfate concentration".

Revised version:

"The increase of the single scattering albedo ($\omega_0$, Fig. 4c) towards the end of the dry season confirms that the aerosol particles during this time are scattering more radiation, not only due to higher BrC presence but also due to other light-scattering aerosol particles".

**REVIEWER**

Lines 568-570 : how do you select and eliminate from the dataset the BB and mineral dust events, and what "extreme event" means (AOD higher than a threshold ?). Please be more precise.

**AUTHORS**

These extreme events were removed from the data used in section 3.4 by using only data within the 90% confidence interval. A comment has been added to the revised version of the manuscript.

**REFERENCES**

Andreae, M. O., Browell, E. V., Garstang, M., Gregory, G. L., Harriss, R. C., Hill, G. F., Jacob, D. J., Pereira, M. C., Sachse, G. W., Setzer, A. W., Dias, P. L. S., Talbot, R. W., Torres, A. L. and Wofsy, S. C.: Biomass-burning emissions and associated haze layers over Amazonia, J. Geophys. Res., 93(D2), 1509, doi:10.1029/JD093iD02p01509, 1988.

Artaxo, P., Martins, J. V., Yamasoe, M. A., Procópio, A. S., Pauliquevis, T. M., Andreae, M. O., Guyon, P., Gatti, L. V. and Cordova Leal, A. M.: Physical and chemical properties of aerosols in the wet and dry seasons in Rondônia, Amazonia, J. Geophys. Res., 107(D20), 8081, doi:10.1029/2001JD000666, 2002.

Formenti, P., Andreae, M. O., Lange, L., Roberts, G., Cafmeyer, J., Rajta, I., Maenhaut, W., Holben, B. N., Artaxo, P. and Lelieveld, J.: Saharan dust in Brazil and Suriname during the Large-Scale Biosphere-Atmosphere Experiment in Amazonia (LBA) - Cooperative LBA Regional Experiment (CLAIRE) in March 1998, J. Geophys. Res. Atmos., 106(D14), 14919–14934, doi:10.1029/2000JD900827, 2001.

Fuzzi, S., Decesari, S., Facchini, M. C., Cavalli, F., Emblico, L., Mircea, M., Andreae, M. O., Trebs, I., Hoffer, A., Guyon, P., Artaxo, P., Rizzo, L. V., Lara, L. L., Pauliquevis, T., Maenhaut, W., Raes, N., Chi, X., Mayol-Bracero, O. L., Soto-García, L. L., Claeys, M., Kourtchev, I., Rissler, J., Swietlicki, E., Tagliavini, E., Schkolnik, G., Falkovich, A. H., Rudich, Y., Fisch, G. and Gatti, L. V.: Overview of the inorganic and organic composition of size-segregated aerosol in Rondônia, Brazil, from the biomass-burning period to the onset of the wet season, J. Geophys. Res. Atmos., 112(1), doi:10.1029/2005JD006741, 2007.

Guyon, P., Graham, B., Beck, J., Boucher, O., Gerasopoulos, E. and Roberts, G. C.: Physical properties and concentration of aerosol particles over the Amazon tropical forest during background and biomass burning conditions, Atmos. Chem. Phys., 3, 951–967, 2003.

Guyon, P., Graham, B., Roberts, G. C., Mayol-Bracero, O. L., Maenhaut, W., Artaxo, P. and Andreae, M. O.: Sources of optically active aerosol particles over the Amazon forest, Atmos. Environ., 38(7), 1039–1051, doi:10.1016/j.atmosenv.2003.10.051, 2004.

Moran-Zuloaga, D., Ditas, F., Walter, D., Saturno, J., Brito, J., Carbone, S., Chi, X., Hrabě de Angelis, I., Baars, H., Godoi, R. H. M., Heese, B., Holanda, B. A., Lavrič, J. V., Martin, S. T., Ming, J., Pöhlker, M., Ruckteschler, N., Su, H., Wang, Y., Wang, Q., Wang, Z., Weber, B., Wolff, S., Artaxo, P., Pöschl, U., Andreae, M. O. and Pöhlker, C.: Long-term study on coarse mode aerosols in the Amazon rain forest with the frequent intrusion of Saharan dust plumes, Atmos. Chem. Phys. Discuss., 1–52, doi:10.5194/acp-2017-1043, 2017.

Prospero, J. M., Glaccum, R. A. and Nees, R. T.: Atmospheric transport of soil dust from Africa to South America, Nature, 289(5798), 570–572, doi:10.1038/289570a0, 1981.

Roberts, G. C., Nenes, A., Seinfeld, J. H. and Andreae, M. O.: Impact of biomass burning on cloud properties in the Amazon Basin, J. Geophys. Res., 108(D2), 4062, doi:10.1029/2001JD000985, 2003.

Talbot, R. W., Andreae, M. O., Berresheim, H., Artaxo, P., Garstang, M., Harriss, R. C., Beecher, K. M. and Li, S. M.: Aerosol chemistry during the wet season in central Amazonia: The influence of long-range transport, J. Geophys. Res., 95(D10), 16955, doi:10.1029/JD095iD10p16955, 1990.

Wang, Q., Saturno, J., Chi, X., Walter, D., Lavric, J. V., Moran-Zuloaga, D., Ditas, F., Pöhlker, C., Brito, J., Carbone, S., Artaxo, P. and Andreae, M. O.: Modeling investigation of light-absorbing aerosols in the Amazon Basin during the wet season, Atmos. Chem. Phys., 16(22), 14775–14794, doi:10.5194/acp-16-14775-2016, 2016.

---

## Author Response (AR1)

[revised manuscript text omitted]

**Figure 4.** Multi-year (2012 – 2017) dry season weekly averages <s>of over the dry season corresponding to</s> (a) frequency of occurrence of BT clusters, *f*, (b) absorption coefficients at 637 nm, ($\sigma_{ap\ 637}$) shown as circles with increasing diameters, the color scale corresponds to the relative BrC contribution to $\sigma_{ap\ 370}$, (c) single scattering albedo at 637 nm, ($\omega_{0\ 637}$), and (d) aerosol optical depth at 550 nm (AOD) for the different domains of interest, DOI1 and DOI2, which cover regions of the South Atlantic Ocean and the southern Amazon, respectively. Boxplots in (c) and (d) represent the median (white segment) and the 25[th] and 75[th] percentiles (lower and upper box edges, respectively).

[Figure]

**Figure 5.** Diel variation of (a, b) median of the accumulation mode particle number concentration, $N_{acc}$, (c, d) median of the absorption coefficient at 637 nm, (e, f) median of the BrC absorption coefficient at 370 nm, (g, h) precipitation rate, and (i, j) median of the equivalent potential temperature. Gray and white backgrounds correspond to the night and day times, respectively. Error bars correspond to the standard error. Please note the different $y$-axis scales .

[Figure]

**Figure 6.** Scatter plot of 2012 – 2017 daily average of BC to CO enhancement ratio, $ER_{BC}$ vs. single scattering albedo at 637 nm, $\omega_{0\ 637}$, with marginal probability density plots (normalized counts in log-scale) for data corresponding to grouped BT clusters.

[Figure]

**Figure 7.** Scattering (a) and absorption (b) coefficient (637 nm) time series measured at the ZF2 and the ATTO sites from 2008 to 2016 (24-h averaged data). Increased scattering and absorption coefficients were observed under the influence of El Niño. (c) High ONI indicates active ENSO periods, shown as red shaded areas.

[Figure]

**Figure 8.** (a) Absorption wavelength dependence (WDA) as function of the OA-to-sulfate mass ratio during high-absorption periods in the dry season. The color scale indicates the $\omega_0$ at 637nm. Gray shaded areas correspond to theoretical WDA for internally mixed BC (light gray), and externally mixed BC (dark gray). The data inside the dashed rectangle in (a) is used in (b) to identify the BT clusters that are more likely to bring BC to the ATTO site.

[Figure]

**Figure 9.** Wavelength dependence of $\mathring{a}_{abs}$ (WDA) for different trajectories in the dry season presented as box and whisker plots (left). The light and dark gray shaded areas correspond to the pure BC and internally mixed BC regimes, respectively. Notches correspond to 1.58 IQR $n^{-\frac{1}{2}}$. If notch ranges do not overlap, the medians are statistically different (95% confidence). The trajectory weighted fire counts for each BT cluster are shown as circles on the right side. The data presented here correspond to 1-h averages.

[Figure]

**Figure 10.** Total absorption at 370 nm (12-h average data) segregated by BC only (gray points) and BrC + BC (brown points). Error bars are equivalent to ± 1 standard error. Long-range transport dust events have been excluded from the analysis.

[Figure]

**Figure 11.** Absorption Ångström exponent ($å_{abs}$) as a function of the BC/OA mass ratio for selected dust events in the wet season. The black line corresponds to a non-linear least squares fit applied to the data ($y = x^{-0.199} \times 0.632$). The light blue lines correspond to the standard error of the fit.

**Response to RC1**

We appreciate the reviewer's comments and suggestions that helped to improve the manuscript. Our responses are presented below, including the original comments from the reviewer, which are presented with gray background.

The paper investigates the occurrence and optical properties of black and brown carbon during a five-year period based on ground-based observations at the ATTO site in the Amazon forest. In particular, the impact of different airmass dynamics and El Nino conditions on the optical properties and relative contribution of black and brown species is investigated. I find the paper well structured and clearly written, and the data presented of great value. In my opinion the paper deserve publication only after minor revisions. Main comments are detailed in the following.

The introduction is quite long and introduces to many concepts. I do not know if it would be better to split it in sub-paragraphs giving a theoretical background on the topic. Anyhow, it is a good state-of-the art of black and brown carbon studies.

**AUTHORS**

We would like to thank the reviewer for the comments related to the introduction and the article in general. We have improved the introduction in order to make it more concise. Please see the revised version of the manuscript.

REVIEWER

Page 14, line 362 : what do you mean with characteristic size distribution ? the average size ? Please be more precise.

**AUTHORS**

We have included number and size distribution plots in the supplementary material to clarify the statements written in the manuscript.

REVIEWER

Section 3.1, page 19, line 482 : you state that dry and wet periods are related to different aerosols influences (biomass burning and dust/sea salt respectively). However different signatures are not present in the temporal absorption angstrom exponent (Fig. 3, panel e). Can you comment on this point ?

**AUTHORS**

The average of hourly mean $\mathring{a}_{abs}$ values measured at the ATTO site during this study was slightly higher during the dry season compared to the wet season, as stated in the original manuscript: "0.94 ± 0.16 compared to a wet season average of 0.91 ± 0.19." We hypothesized a more pronounced seasonality due to the larger occurrence of fires in the dry season bringing BrC-rich aerosol. However, we have found that these conditions occurred episodically rather than seasonally, with increased $\mathring{a}_{abs}$ only when ATTO was under strong influence of likely close-by biomass burning.

The following comment was added to section 3.1 of the revised manuscript:

"It was found that the $\mathring{a}_{abs}$ only increased significantly during episodes of hours or days, typically caused by nearby burning during the dry season".

Regarding the mineral dust effect on $\mathring{a}_{abs}$, no influence was found during the dust periods, as stated in the manuscript: "no effect on $\mathring{a}_{abs}$ was observed due to the presence of dust, most likely due to a size effect, given that absorption coefficients were measured only for sub-micron aerosol particles after May 2014".

REVIEWER

Line 492-526 : probably this part can be moved in a single paragraph focusing on the MAC

**AUTHORS**

Point taken; in the new version of the manuscript the MAC discussion is included in a new section.

REVIEWER

Line 529-530 : many time the authors state, but do not prove, that the dry season is affected by BB particles. Is this assumption made based on previous studies at the site? The same for the dust influence during the wet season. I think this point should be better addressed in the paper before analyzing in more detail the optical properties of the different aerosol types in different periods. This was the only part that I found not clear at all in the paper.

**AUTHORS**

Point taken. The influences of biomass burning and mineral dust over the Amazon rain forest are indeed well documented in the literature, therefore we have included the following references in the introduction of the revised manuscript:

Biomass burning influence during the dry season:

(Andreae et al., 1988; Artaxo et al., 2002; Fuzzi et al., 2007; Guyon et al., 2003; Roberts et al., 2003)

Saharan dust influence during the wet season:

(Formenti et al., 2001; Guyon et al., 2004; Moran-Zuloaga et al., 2017; Prospero et al., 1981; Talbot et al., 1990; Wang et al., 2016), already mentioned in the original manuscript.

REVIEWER

Line 565 : you state that is the contribution of sulfate that increases scattering. Why not the mixing with other compounds or species ?

**AUTHORS**

We agree with the reviewer and have modified the manuscript accordingly.

Original version:

> "In terms of the single scattering albedo ($\omega_0$, Fig. 4c), its increase towards the end of the dry season confirms that the aerosol particles during this time are scattering more radiation, not only due to higher BrC presence but also due to an increased sulfate concentration".

Revised version:

> "The increase of the single scattering albedo ($\omega_0$, Fig. 4c) towards the end of the dry season confirms that the aerosol particles during this time are scattering more radiation, not only due to higher BrC presence but also due to other light-scattering aerosol particles".

**REVIEWER**

Lines 568-570 : how do you select and eliminate from the dataset the BB and mineral dust events, and what "extreme event" means (AOD higher than a threshold ?). Please be more precise.

**AUTHORS**

These extreme events were removed from the data used in section 3.4 by using only data within the 90% confidence interval. A comment has been added to the revised version of the manuscript.

**Response to RC2**

We appreciate the reviewer's comments and suggestions, which helped to improve the manuscript. Our responses are presented below, including the original comments from the reviewer, which are presented with gray background.

**GENERAL**

The paper presents analyses of aerosol optical properties measured at the ATTO site in Amazonia during several years. The authors have measured scattering and absorption coefficients, refractory BC (rBC), aerosol chemical composition as well as several supporting parameters. The wavelength dependence of absorption was used for estimating the contributions of black and brown carbon to light absorption, contributions of geographical source areas were estimated using transport analyses, mass absorption coefficients (MAC) by comparing independent absorption and rBC measurements. All this is important and valuable. The paper is mainly well written so I can recommed its publication after some clarifications and revision.

REVIEWER

The main point that bothers me is the way the contribution of brown carbon is calculated. It is the core of the paper so it should be presented clearly. I'll show the problem in the detailed comments.

**AUTHORS**

The details about how the BrC contribution was calculated are described later in this document when addressing the detailed comments of the reviewer.

REVIEWER

Another point that I miss is the size distributions of rBC. They were measured with the SP2 and used for MAC calculations but not shown anywhere. Why? The geometric mean diameters and geometric standard deviations of BC are useful and valuble as such for modeling purposes. Did they vary seasonally and with source areas? The size distríbution

can also give hints of whether part of the BC remained undetected which would definitely affect the calculated MAC values, another important point in this paper. There should be some uncertainty analysis of the MAC.

**AUTHORS**

Agreed – the rBC size distributions and the instrumental counting efficiency are elemental for the calculations presented in this manuscript. Its revised version includes selected rBC size distributions (Fig. S5) and an estimate of the MAC uncertainty. These data will be the subject of a more detailed analysis in a future publication.

The rBC core size distributions included in the revised version are the following:

[Figure]

**Figure S5**. Refractory black carbon mass size distributions observed at the ATTO site on different characteristic days during the wet (blue dots) and dry (red dots) seasons in 2014. The right panel shows a zoom into the wet season size distribution.

**REVIEWER**

How about coatings? They can be obtained from the SP2 but not presented, why? It would be valuable for the analysis of absorption enhancement.

**AUTHORS**

The coating information is indeed relevant. However, a detailed SP2 data analysis would be beyond the scope of this manuscript and is the subject of a future study.

**REVIEWER**

**DETAILED COMMENTS**

L112 "(Womack et al., ref needed)." Write the ref.

**AUTHORS**

We apologize for the missing reference. It is included in the revised version.

**REVIEWER**

L221-227. I assume you corrected also the Aurora 3000 data for truncation, did you?

**AUTHORS**

Yes, the Aurora 3000 data was corrected according to Müller et al. (2011). This is clarified in the revised version of the manuscript.

**REVIEWER**

L234 "... instrument is able to provide absorption coefficients with a time resolution of 5 min." The time resolution of the MAAP can be set not only to 5 min. Reword the sentence.

**AUTHORS**

Agreed – the corrected text reads:

Original version:

> "... instrument is able to provide absorption coefficients with a time resolution of 5 min."

Revised version:

> "... the instrument is able to provide absorption coefficients. The instrument was set up to provide data at 1-min resolution".

REVIEWER

L 303-304 "... . The 8-channel SP2 rBC mass measurement was underestimated by a factor of 5 % ..."

How was this 5% obtained?

Another thing is, how could you estimate missing BC if it were outside the size range detected by the SP2? In biomass burning smoke BC could be attached to larger particles as well.

**AUTHORS**

Agreed. The SP2 rBC counting efficiency drops significantly for particles smaller than 80 nm diameter. We have found the mentioned scaling factor by comparing SP2 counts vs. CPC size resolved counts of fullerene particles. Similar offset values have been found in the literature when comparing SP2 counts vs. condensation particle counters (Liu et al., 2017) or by fitting the SP2 rBC number or mass size distributions and calculating the missing mass fraction for the smaller particles (Wang et al., 2014). This estimation introduces uncertainties in the SP2 results, which in total, including those from the mass calibration, reach around 25 % uncertainty (Wang et al., 2014).

Regarding the BC attached to larger particles, we are aware of this kind of mixing and it can also occur over the Amazon rain forest. However, we have found no significant difference in the MAAP vs. SP2 offset when measuring particles below 1 µm (using a PM1 cyclone) vs. total particle measurements, which suggests that the fraction of BC attached to large particles is rather low.

We addressed the issue brought up by the referee by replacing the original section on page 12, line 303-304:

> "The 8-channel SP2 rBC mass measurement was underestimated by a factor of 5 %, related to the size-dependent detection efficiency of the instrument, which is below 100 % in the 80 to 150 nm diameter range. Therefore, a scaling factor of 1.05 was applied to rBC mass concentration data to account for this systematic error".

by the following revised version:

> "The 8-channel SP2 rBC size-dependent counting efficiency was obtained by comparing the counts of fullerene particles measured by the SP2 and a condensation particle counter (CPC). This way, an underestimation factor of 5 % was found to affect

SP2 rBC mass measurements and a scaling factor of 1.05 was applied to the data to account for this systematic error. Similar underestimation factors have been previously reported (Liu et al., 2017; Wang et al., 2014)".

**REVIEWER**

L356-372, calculation of BrC. There is a problem here. I rewrite the equations and show it.

$$WDA = \mathring{a}_{abs370-950} - \mathring{a}_{abs660-950} \tag{3}$$

$$BC\mathring{a}_{abs370-950} = \mathring{a}_{abs660-950} + WDA \tag{4}$$

$$BC\sigma_{ap370} = \sigma_{ap950} \times \left(\frac{370}{950}\right)^{-BC\mathring{a}_{abs370-950}} \tag{5}$$

$$BrC\sigma_{ap370} = \sigma_{ap370} - BC\sigma_{ap370} \tag{6}$$

Insert (3) to (4)

$$\Rightarrow BC\mathring{a}_{abs370-950} = \mathring{a}_{abs660-950} + WDA = \mathring{a}_{abs660-950} + \mathring{a}_{abs370-950} - \mathring{a}_{abs660-950} = \mathring{a}_{abs370-950}$$

Insert the result to (5)

$$\Rightarrow BC\sigma_{ap370} = \sigma_{ap950} \times \left(\frac{370}{950}\right)^{-BC\mathring{a}_{abs370-950}} = \sigma_{ap950} \times \left(\frac{370}{950}\right)^{-\mathring{a}_{abs370-950}} = \sigma_{ap370}$$

where the last step comes from applying (1) on line 342.

Insert finally the result to (6) and you get that $BrC\sigma_{ap}370 = 0$.
This cannot be the idea, BrC being always zero. Rewrite the equations.

**AUTHORS**

We understand that the way the equations were written in the manuscript could be misleading because it was not clear whether the absorption Ångström exponents were modeled or measured. We have fixed it by including superscripts that indicate when $\mathring{a}_{abs}$ was modeled (superscript "BC") or measured (no superscript).

The revised version is below:

$$\sigma_{ap\,370}^{BC} = \sigma_{ap\,950} \times \left(\frac{370}{950}\right)^{-\mathring{a}_{abs\,370-950}^{BC}} \quad , \tag{4}$$

$$\sigma_{ap\,370}^{BrC} = \sigma_{ap\,370} - \sigma_{ap\,370}^{BC} \quad , \tag{5}$$

where $\mathring{a}_{abs\,370-950}^{BC}$ is obtained from the Mie model calculations.

Further on the same issue. On L361-362 it is written " Calculated BC WDA thresholds, presented in Fig. S5, were compared to the ambient data in order to retrieve the BrC contribution to light absorption. "

What do you mean by thresholds? Do you mean that if WDA is larger than a threshold then this is due to BrC? This is not clear at all. Looking at Fig S5 does not explain me, what these thresholds might be. And what is the reasoning for claiming that exceeding a threshold for WDA is due to BrC?

This is much too descriptive way to explain how you calculated BrC. Give more details so that other people can us the same method and evaluate it. Recently many people have started using the so-called "Aethalometer model" (Sandradewi et al., 2008) and it would be good if the model presented in this work could be compared with it.

**AUTHORS**

Agreed. The WDA thresholds used in this study were the 25[th] and 75[th] percentiles of the modeled BC wavelength dependence and they are shown as dashed lines in Fig. S6 (Fig. S5 in the original version). These percentiles were calculated using data of particles from 100 to 275 nm diameter. As can be seen in Fig. S6, the inter-quartile range comprises internally mixed particles of the following diameters: 125, 150, 225, and 275 nm with coating thickness to core size ratio > 0.3, and part of the particles with 175 and 200 nm diameter. This range also includes most of the externally mixed particles in the size range from 100 to 225 nm diameter. When the 75[th] percentile threshold is exceeded, the particles are considered to include BrC additionally to BC. The sensitivity of this model was tested by changing the core size diameters and the refractive index of the coating material and the results were expressed as "relative overestimation" of the BrC carbon contribution to $\sigma_{370}$, as shown in Table S2.

We addressed the issue brought up by the referee by replacing the original section on page 14, line 361-362:

> "Calculated BC WDA thresholds, presented in Fig. S5, were compared to the ambient data in order to retrieve the BrC contribution to light absorption."

by the following revised version:

"Calculated BC WDA thresholds (25$^{th}$ and 75$^{th}$ percentiles), shown in Fig. S6, were compared to the ambient data in order to identify BrC influenced periods. For a general analysis, data with WDA lower than the 75th percentile were considered to be in the *BC-only* regime. The presence of BrC, additionally to BC, occurred when the modeled BC absorption at 370 nm was exceeded."

**REVIEWER**

L373-377, still about the same issue. How did you get the uncertainties? Give formulas. Writing that " The relative overestimation of the BrC contribution obtained by using different BC core sizes and different refractive indices in the Mie model calculations can be found in Table S2." is simply too qualitative an not understandable. People have to be able to reproduce the result.

**AUTHORS**

The results presented in Table S2 were actually a sensitivity evaluation of the model and not a real uncertainty analysis. Basically, the WDA thresholds were recalculated by changing different parameters in the model. This clarification is included in the revised version of the manuscript.

Revised version:

"A sensitivity study of this model was done by changing the refractive indices and the core size of the model input. These results are presented in Table S2 as relative overestimation of the BrC contribution to $\sigma_{ap\ 370}$".

**REVIEWER**

L472-473 " Rizzo et al. (2013), however, pointed out that this relationship is only evident for surface and volume
mean diameters and was not clearly valid between åsca and count mean diameters. "
Also Virkkula et al.: ACP, 11, 4445–4468, 2011 found the same.

**AUTHORS**

We thank the reviewer for pointing out about this reference and will include it in the revised version.

REVIEWER

L568 -> Where do you get the equivalent potential temperature from? Model, I assume, present in methods then.

**AUTHORS**

The potential temperature, $\theta_e$, was calculated according to Stull (1988), as follows:

$$\theta_e = T_e \cdot \left(\frac{p_0}{p}\right)^{\frac{R}{c_p}} \approx \left(T + \frac{L_v}{c_p}r\right)\left(\frac{p_0}{p}\right)^{\frac{R}{c_p}}$$

where

$T$: Air temperature.

$p$: Atmospheric pressure.

$p_0$: Reference pressure (1000 hPa).

$R$: Gas constant for air.

$c_p$: Specific heat constant for pressure.

$L_v$: Latent heat of evaporation.

$r$: Water vapor mixing ratio

The cited reference is included in the revised version of the manuscript.

REVIEWER

L672 Do you have a clear definition for "BC-only regime"?

**AUTHORS**

We thank the reviewer for asking about this because it was not clear in the submitted version of the manuscript. A comment has been included in the revised version, section 2.3:

> "For a general analysis, data with WDA lower than the 75[th] percentile were considered to be in the *BC-only* regime".